# DivBO: Diversity-aware CASH for Ensemble Learning

**Yu Shen**[1], **Yupeng Lu**[1], **Yang Li**[4], **Yaofeng Tu**[3], **Wentao Zhang**[5][6], **Bin Cui**[1][2]

[1]Key Lab of High Confidence Software Technologies, Peking University, China
[2]Institute of Computational Social Science, Peking University (Qingdao), China
[3] ZTE Corporation, China [4] Data Platform, TEG, Tencent Inc., China
[5] Mila - Québec AI Institute [6]HEC, Montréal, Canada
[1]{shenyu, xinkelyp, bin.cui}@pku.edu.cn, [2] tu.yaofeng@zte.com.cn
[3]thomasyngli@tencent.com [4]wentao.zhang@mila.quebec

## Abstract

The Combined Algorithm Selection and Hyperparameters optimization (CASH) problem is one of the fundamental problems in Automated Machine Learning (AutoML). Motivated by the success of ensemble learning, recent AutoML systems build post-hoc ensembles to output the final predictions instead of using the best single learner. However, while most CASH methods focus on searching for a single learner with the best performance, they neglect the diversity among base learners (i.e., they may suggest similar configurations to previously evaluated ones), which is also a crucial consideration when building an ensemble. To tackle this issue and further enhance the ensemble performance, we propose DivBO, a diversity-aware framework to inject explicit search of diversity into the CASH problems. In the framework, we propose to use a diversity surrogate to predict the pair-wise diversity of two unseen configurations. Furthermore, we introduce a temporary pool and a weighted acquisition function to guide the search of both performance and diversity based on Bayesian optimization. Empirical results on 15 public datasets show that DivBO achieves the best average ranks (1.82 and 1.73) on both validation and test errors among 10 compared methods, including post-hoc designs in recent AutoML systems and state-of-the-art baselines for ensemble learning on CASH problems.

## 1 Introduction

In recent years, machine learning has made great strides in various application areas, e.g., computer vision [14, 12], recommendation [36, 37], etc. However, it's often knowledge-intensive to develop customized solutions with promising performance, as the process includes selecting proper ML algorithms and tuning the hyperparameters. To reduce the barrier and facilitate the deployment of machine learning applications, the AutoML community raises the Combined Algorithm Selection and Hyperparameters optimization (CASH) problem [39] and proposes several methods [33, 18, 15] to automate the optimization.

While most AutoML methods [35, 17, 2] for CASH focus on searching for the optimal performance of a single learner, it's widely acknowledged that ensembles of promising learners often outperform single ones [44, 3]. For example, He et al. [14] won first place in ILSRVC 2015 with an average of several learners. And ensemble strategies can be frequently found in the top solutions of Kaggle competitions [16, 4]. Motivated by those achievements, recent AutoML systems (e.g., Auto-sklearn [11], Auto-Pytorch [50], VolcanoML [25]) build post-hoc ensembles based on all base learners from the entire optimization process and show better empirical results than using the best single learner.

Despite the effectiveness of those post-hoc ensemble designs, the target of CASH methods is inconsistent with that of ensemble learning. In other words, a good ensemble should contain a pool of

base learners that are both well-performing and diverse with each other [43, 44], while most CASH methods only aim at searching for the best-performing configuration of learners. Figure 1 shows the diversity of the suggested learners from Auto-sklearn and VolcanoML after 200 iterations. We define the pair-wise diversity of two learners as the average regularized Euclidean distance of their probabilistic predictions which ranges from 0 to 1 (See Equation 3). And we show the minimum diversity of a suggested learner with other learners from the ensemble built at the $200^{th}$ iteration. While most diversity values of the two AutoML systems are around 0.05, they suggest learners whose predictions are often quite similar to one of the learners obtained from previous iterations. It implies a lack of diversity in previous post-hoc ensemble designs, and it also indicates that the performance of post-hoc ensembles might be further improved if diversity is also taken into consideration when suggesting new configurations to evaluate. However, the search of diversity is non-trivial. A simple diversity-inducing algorithm would degenerate the pool of learners, e.g., diversity can be easily increased by predicting all samples for the classes that are wrong and different from previous learners. Therefore, how to guide the search for both performance and diversity simultaneously in recent CASH algorithms is still an open question.

In this paper, we propose DivBO, a new algorithm framework that combines Bayesian optimization (BO) with an explicit search of diversity for classification. The contributions are summarized as:

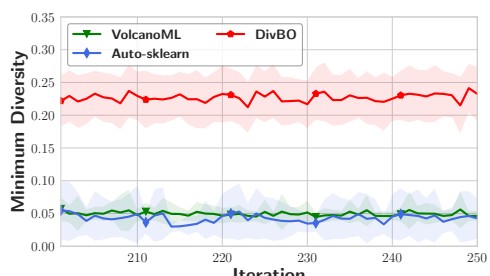

Figure 1: Minimum diversity of suggested learners after 200 iterations on quake. (20 runs)

1. To the best of our knowledge, this work is the first to enhance the ensemble performance of CASH methods by considering the diversity of suggested configurations.

2. To inject the search of diversity into CASH problems, we design 1) the diversity surrogate to predict the pair-wise diversity given two unseen configurations and 2) a BO-based framework that automatically balances the predictive performance and diversity of suggestions.

3. Empirical results show that the diversity surrogate achieves a better correlation with the ground-truth results compared with the performance surrogate used in BO. Compared with recent AutoML systems, DivBO suggests configurations that are significantly more diverse with those in previous iterations (the red line in Figure 1). In addition, results on 15 public datasets show that DivBO achieves the best average rank (1.82 and 1.73) on both validation and test errors among 10 methods, including post-hoc designs in recent AutoML systems and state-of-the-art methods for ensemble learning on CASH problems.

## 2 Preliminary and Related Work

In this section, we review the related work and introduce the preliminary for our proposed method.

**Combined algorithm selection and hyperparameters (CASH).** We first introduce the basic notations for the CASH problem. Let $\mathcal{A}$ be the set of $K$ candidate algorithms $\mathcal{A} = \{A^1, ..., A^K\}$. Each algorithm $A^i$ has its corresponding hyperparameter space $\Lambda_{A^i}$. Given dataset $\mathcal{D} = \{\mathcal{D}_{train}, \mathcal{D}_{val}\}$ of a learning problem, the CASH problem aims to minimize the validation metric $\mathcal{L}$ by searching for the best joint configuration $x^* = (A^*, \lambda^*)$ trained on the training set, where $A^*$ is the best algorithm and $\lambda^*$ is its corresponding hyperparameters. For brevity, we use $T_{(a,\lambda)}$ to denote the learner constructed by the joint configuration $x = (a, \lambda)$ and trained on $\mathcal{D}_{train}$. The optimization objective can be formulated as:

$$\min_{a \in \mathcal{A}, \lambda \in \Lambda_a} \mathcal{L}(T_{(a,\lambda)}, \mathcal{D}_{val}). \qquad (1)$$

The CASH problem is first introduced by Auto-WEKA [39] and solved by applying Bayesian optimization (BO) [35, 17, 2] on the entire search space. Each BO iteration follows three steps: 1) BO fits a probabilistic surrogate model based on observations $D = \{(x_1, y_1), ..., (x_{n-1}, y_{n-1})\}$, in which $x_i$ is the $i^{th}$ evaluated joint configuration, and $y_i$ is its corresponding observed performance; 2) BO selects the most promising configuration $x_n$ by maximizing the acquisition function to balance exploration and exploitation; 3) BO evaluates the configuration $x_n$ (i.e., train the learner and obtain

its validation performance), and augment the observations $D$. To avoid confusion, we refer to the surrogate of BO as the performance surrogate $M_{perf}$ in the following section.

In addition to BO, TPOT [31] proposes to use genetic programming to solve the CASH problem. The ADMM-based method [29] decomposes the problem into sub-problems and solves them using ADMM [5]. Rising bandit [23] continuously eliminates badly performing algorithms during optimization. FLAML [41] suggests configurations based on estimated costs of improvement.

**Ensemble-oriented CASH.** Different from the original CASH, ensemble-oriented CASH aims to find an ensemble of $m$ learners that minimizes the validation metric $\mathcal{L}$, which is:

$$\min_{a_1,...,a_m \in \mathcal{A}, \lambda_1 \in \Lambda_{a_1},...,\lambda_m \in \Lambda_{a_m}} \mathcal{L}(Ensemble(T_{(a_1,\lambda_1)},...,T_{(a_m,\lambda_m)}), \mathcal{D}_{val}). \tag{2}$$

To solve the problem, ensemble optimization [22] directly optimizes the target based on BO by using an ensemble pool with a fixed size. This strategy suffers from instability due to the risk of adding a bad configuration during optimization. Neural ensemble search (NES) [43] is proposed based on the regularized evolutionary algorithm [34]. The local search design in NES works well on low-dimensional problems like neural architecture search, but may encounter a bottleneck when applied to solve high-dimensional CASH problems.

Rather than directly optimizing on Equation 2, recent AutoML systems (e.g., Auto-sklearn [11], VolcanoML [25, 26], Auto-Pytorch [50]) adopt a specific ensemble strategy on the observation history after solving Equation 1, which we refer to as the **post-hoc ensemble** designs. While the optimization target is inconsistent, as shown in Section 1, there's still space to improve the performance of post-hoc ensembles. Inspired by the theoretical study that diversity can help improve the performance of the ensemble [21, 3, 13, 48], we follow the post-hoc ensemble designs and focus on generating a promising pool of learners for the final ensemble strategy.

**Ensemble selection.** Ensemble strategies are methods that combine the predictions given a pool of learners, which are orthogonal to the direction of our method (i.e., suggesting a diverse pool of learners). Among different ensemble strategies (e.g., Bagging [9], Boosting [9, 30], Stacking [6]), we adopt the ensemble selection [7], which works empirically well with AutoML as shown in previous study [11, 25, 43]. In short, ensemble selection starts from an empty ensemble and iteratively adds models from the pool with replacement to maximize the ensemble validation performance (with uniform weights). The pseudo-code is provided in Appendix A.1.

## 3 Diversity-aware Bayesian optimization (DivBO)

In this section, we present DivBO — our proposed diversity-aware CASH method for ensemble learning based on Bayesian optimization (BO). To inject the search of diversity into BO, we will answer the following two questions: 1) how to measure diversity and model the diversity of two unseen configurations, and 2) how to suggest configurations that are both well-performing and diverse with potential ones in the final ensemble.

### 3.1 Diversity Surrogate

The existing diversity measures can be generally divided into pair-wise and nonpair-wise measures. The nonpair-wise diversity [47] directly measures a set of learners in the ensemble. While the ensemble may change during each iteration, it's difficult to model the diversity of an ensemble with a candidate configuration and multiple learners. Therefore, we use the pair-wise measures to simply learn the diversity of two given configurations. To this end, we follow the definition in previous research [44] that explicitly improves the diversity of neural networks and shows satisfactory empirical results. Let $x_i$ denote as the joint configuration of algorithm $a_i$ and hyperparameters $\lambda_i$. The diversity between two configurations $(x_i, x_j)$ is the average Euclidean distance of class probabilities predicted on the validation set [44], which is:

$$Div(x_i, x_j) = \frac{\sqrt{2}}{2} \frac{1}{|\mathcal{D}_{val}|} \sum_{s \in \mathcal{D}_{val}} ||T_{x_i}(s) - T_{x_j}(s)||_2, \tag{3}$$

where $\mathcal{D}_{val}$ is the validation set, $T_{x_i}$ is the learner corresponding to configuration $x_i$ and fitted on the training set, and $T_{x_i}(s)$ is the predictive class probability on sample $s$. Obviously, the relationship

between the diversity and configuration pair $(x_i, x_j)$ is also a black-box function. Therefore, inspired by Bayesian optimization, we apply another surrogate to model this relationship. This surrogate, namely the diversity surrogate, takes a configuration pair as input and outputs the predictive mean and variance of the pair-wise diversity.

**Fitting.** During each iteration, DivBO generates the training set for the diversity surrogate $M_{div}$ by computing the diversity value of each pair of the observed configurations. Note that, $(x_i, x_j)$ and $(x_j, x_i)$ lead to the same diversity value, but the training set should include both of them to ensure symmetry. This leads to about $|D|^2$ training samples, where $|D|$ is the number of observations. To avoid fetching validation predictions by retraining, we store those predictions for all observations during optimization, which is a common trick applied in previous methods [11, 22, 43].

**Implementation.** Concretely, the diversity surrogate $M_{div}$ in DivBO is an ensemble of several LightGBM models [19]. This implementation has the following two advantages: 1) The diversity surrogate gives predictions with uncertainty to balance exploration and exploitation. Concretely, the predictive mean and variance of the surrogate is obtained by computing the mean and variance of outputs generated by different LightGBM models. 2) The time complexity of fitting the surrogate is relatively low. The cost of training a LightGBM model is $O(|D|^2 \log |D|)$, which is lower than $O(|D|^3)$ when fitting a Gaussian Process as the performance surrogate. In Section 4, we will further compare different implementations of diversity surrogates on their ability to fit the relationship.

### 3.2 Diversity-aware Framework

Based on the diversity surrogate, we propose a diversity-aware framework to suggest configurations that are both well-performing and diverse. Before stepping into the design, consider a straightforward strategy that we choose the most diverse configuration with the observation history during each iteration. This simple strategy has two obvious drawbacks: 1) First, there is no need to suggest configurations that are diverse from badly-performing ones in the observation history—we only expect the promising base learners in the final ensemble to be diverse with each other; 2) The diversity can be easily increased by suggesting a learner that predicts all samples for the classes that are wrong and different from previous learners. In this extreme case, the learner can not help improve the performance of the final ensemble. Therefore, how to control the diversity of the suggested configurations is non-trivial. In the following, we explain how DivBO tackles the two drawbacks.

**Suggesting diverse configurations.** To tackle the first drawback, since which learners will appear in the final ensemble is unknown during optimization, DivBO proposes to use a **temporary pool** $\mathcal{P}$ to collect base learners that will probably be selected into the final ensemble. Concretely, the pool is built by applying ensemble selection to the observation history. When the optimization ends, this temporary pool is exactly the output of a post-hoc ensemble strategy (e.g., auto-sklearn[11]) that ends at the previous iteration, and thus the configurations in the temporary pool can be regarded as the potential ones in the final ensemble. Then, we define the diversity acquisition function $\alpha_{div}$ of each unseen configuration $x$ as its predicted diversity value with the most similar configuration in the temporary pool:

$$\alpha_{div}(x) = \frac{1}{N} \sum_{n=1}^{N} \min_{\theta \in \mathcal{P}} M_{div}^n(\theta, x), \tag{4}$$

where $\mathcal{P}$ is the temporary configuration pool and $M_{div}$ is the diversity surrogate. $M_{div}^n(\theta, x)$ is the $n^{th}$ sampled value from the output distribution of $M_{div}$ given the pair $(\theta, x)$, and the final acquisition function is the average of $N$ minimums via sampling. Note that, using the minimum diversity with learners in the temporary pool is more appropriate than the mean diversity. In extreme cases, when the algorithm may suggest a useless configuration that is exactly the same as a previously evaluated one, the minimum diversity is penalized to zero while the mean diversity is still larger than zero. By maximizing Equation 4, DivBO is able to suggest configurations that are diverse from other potential ones in the final ensemble during optimization.

**Combining predictive performance and diversity.** As mentioned in the second drawback, optimizing diversity alone degenerates the predictive performance of the suggested configurations. DivBO tackles this issue by combining both the performance and diversity acquisition functions with a saturating weight. Since the performance and diversity acquisition values are of different scales, we propose to use the sum of ranking values instead of directly adding the output values. During each

---

**Algorithm 1:** Algorithm framework of DivBO.

---

**Input:** The search budget $\mathcal{B}$, the architecture search space $\mathcal{X}$, the ensemble size $E$, the training and validation set $\mathcal{D}_{train}, \mathcal{D}_{val}$.

**1** Initialize observations as $D = \varnothing$;

**2 while** *budget $\mathcal{B}$ does not exhaust* **do**

**3**    **if** $|D| < 5$ **then**

**4**      Suggest a random configuration $\tilde{x} \in \mathcal{X}$;

**5**    **else**

**6**      Fit performance surrogate $M_{perf}$ and diversity surrogate $M_{div}$ based on observations $D$;

**7**      Build a temporary pool of configurations as
      $\mathcal{P} = \{\theta_1, ..., \theta_E\} = EnsembleSelection(D, \mathcal{D}_{val}, E)$;

**8**      Compute the ranks of sampled configurations $R_{perf}$ and $R_{div}$ based on the performance
      and diversity surrogates $M_{perf}, M_{div}$ and the temporary pool $\mathcal{P}$;

**9**      Suggest a configuration $\tilde{x} = \underset{x \in \mathcal{X}}{\arg\min}\, \alpha(x)$ based on Equation 5;

**10**    Build and train the learner $f_{\tilde{x}}$ on $\mathcal{D}_{train}$ and evaluate its performance on $\mathcal{D}_{val}$ as $\tilde{y}$;

**11**    Update the observations $D = D \cup \{(\tilde{x}, \tilde{y})\}$;

**12** Generate a pool of base configurations $\{\theta_1, ..., \theta_E\} = EnsembleSelection(D, \mathcal{D}_{val}, E)$;

**13 return** the final ensemble $Ensemble(T_{\theta_1}, ..., T_{\theta_E})$.

---

BO iteration, we sample sufficient configurations by random sampling from the entire space and local sampling on well-performed observed configurations [17]. Then we calculate the performance and diversity acquisition value for each sampled configuration. Based on these values, we further rank the sampled configurations and obtain the ranking value of $x_i$ as $R_{perf}(x_i)$ and $R_{div}(x_i)$ for the performance and diversity acquisition function, respectively. Finally, given a configuration $x_i$, we define the weighted acquisition function $\alpha$ for DivBO as:

$$\alpha(x_i) = R_{perf}(x_i) + wR_{div}(x_i), \quad w = \beta(sigmoid(\tau t) - 0.5), \quad (5)$$

where $w$ is the weight for the diversity acquisition, $t$ is the current number of BO iterations, $\beta$ and $\tau$ are two hyperparameters. To match the intuition that the optimization should focus on performance in the beginning and gradually shift its attention to diversity, DivBO applies a saturating weight, where $w$ ranges from $[0, \beta)$ and $\tau$ controls the speed of approaching saturation. The goal of DivBO is to suggest configurations that **minimize** the function in Equation 5.

Algorithm 1 illustrates the procedure of DivBO. During each iteration after initialization, DivBO 1) fits the performance and diversity surrogates based on observations (Line 6); 2) builds a temporary configuration pool by applying ensemble selection on the observation history (Line 7); 3) samples candidate configurations and compute their ranking values (Line 8); 4) suggests a configuration that minimizes the combined ranking value in Equation 5 (Line 9); 5) evaluates the suggested configuration on the validation set and then updates the observations (Lines 10-11).

### 3.3 Discussion

In this section, we provide the discussion on DivBO as follows:

**Time complexity.** As mentioned in Section 3.1, the time complexity of fitting the diversity surrogate is $O(|D|^2 \log|D|)$, and the complexity of building a temporary pool is $O(|\mathcal{D}_{val}||D|)$, where $|\mathcal{D}_{val}|$ is the number of validation data samples, and $|D|$ is the number of observations. For large datasets, we can prepare a constant subset of validation samples for ensemble selection. Therefore, the time complexity depends on the choice of performance surrogate in BO. Concretely, the complexity of DivBO for each iteration is $O(|D|^3)$ when using the Gaussian Process [35] and $O(|D|^2 \log|D|)$ when using the probabilistic random forest [17].

**Extension.** As an abstract algorithm framework, the components of DivBO can be replaced to meet different requirements. Though this paper focuses on CASH problems for classification, DivBO can also be applied to regression problems by defining a new diversity function based on regression predictions instead of Equation 3. In addition, DivBO is independent of the choice of

performance surrogate for Bayesian optimization, i.e., the performance surrogate can be replaced with state-of-the-art ones proposed for specific scenarios.

**Foundation.** Like the diversity-driven methods in other scenarios [44, 32], DivBO does not target at optimizing Equation 2 directly. The foundation of its effectiveness lies in the claim that it is beneficial for ensembles to not only have promising base learners, but also more diversity in their predictions. The claim has been studied by extensive theoretical work [21, 3, 13, 48, 20], and please refer to previous work for more details. In the following section, we will empirically show that the ensemble generated by DivBO outperforms the state-of-the-art methods in real-world CASH problems.

**Limitations.** The use of ensemble leads to higher inference latency than using the single best learner (approximately K times where K is the number of learners in the ensemble). This latency can be reduced with the aid of parallel computing if we have sufficient computational resources; As ensemble selection is fitted on the validation set, there's a risk of overfitting on the test set for small datasets; DivBO using Equation 3 as the diversity function can not directly support algorithms that only predict class labels (e.g., SVC). Though DivBO still works by converting the outputs to class probability (like [1, 0, ...]), other diversity functions can be developed to support those algorithms better.

## 4 Experiments

In this section, we evaluate our proposed method on real-world CASH problems using public datasets. In the following, we list three main insights that we will investigate: 1) The diversity surrogate in DivBO can predict the diversity value of two unseen configurations well. 2) The DivBO framework outperforms the post-hoc designs used in recent AutoML systems and other competitive baselines for ensemble learning, in terms of both validation and test performance. 3) The base learners in the ensemble given by DivBO show similar average performance but enjoy higher diversity than those from other post-hoc designs.

### 4.1 Experiment Setup

**Baselines.** We compare the proposed DivBO with the following eight baselines — *Three CASH methods:* 1) Random search (RS) [1]; 2) Bayesian optimization (BO) [17]; 3) Rising Bandit (RB) [23]; — *Two AutoML methods proposed for ensemble learning:* 4) Ensemble optimization (EO) [22]; 5) Neural ensemble search (NES) [43]; — *Three post-hoc designs:* 6) Random search with post-hoc ensemble (RS-ES); 7) Bayesian optimization with post-hoc ensemble (BO-ES) [11]: the default strategy in Auto-sklearn; 8) Rising bandit with post-hoc ensemble (RB-ES) [25]: the default strategy in VolcanoML. While for DivBO, we also implement the variant without post-hoc ensemble, which we denote as "DivBO-".

**Datasets and search space.** While recent AutoML systems differ in both search space and algorithm, to make a fair comparison of algorithms, we conduct the experiments on the same search space. Concretely, we slightly modify the search space of the AutoML system VolcanoML [25]. The search space contains 100 hyperparameters in total, and the details of algorithms and feature engineering hyperparameters are provided in Appendix A.3. In addition, we use 15 public classification datasets that are collected from OpenML [40], whose number of samples ranges from 2k to 20k. More details about the datasets are provided in Appendix A.2.

**Basic settings.** Each dataset is split into three sets, which are the training (60%), validation (20%), and test (20%) sets. To evaluate the diversity surrogate, we use the Kendall-tau rank correlation as the metric since we only care about the ranking relationship between two pairs during optimization. For comparison with other baselines on CASH problems, we report the best-observed validation error during optimization and the final test error. While it takes a different amount of time to evaluate the same configuration on different datasets, we use the evaluation iterations as the unit of budget. Following VolcanoML [25] where each baseline evaluates approximately 250 configurations, we set the number of maximum iterations to 250. The evaluation of each method on each dataset is repeated 10 times, and we report the mean±std. result by default. The hyperparameters $\beta$ and $\tau$ are set to 0.05 and 0.2 in DivBO. We provide more implementation details for other baselines and the sensitivity analysis in Appendix A.4 and A.5, respectively.

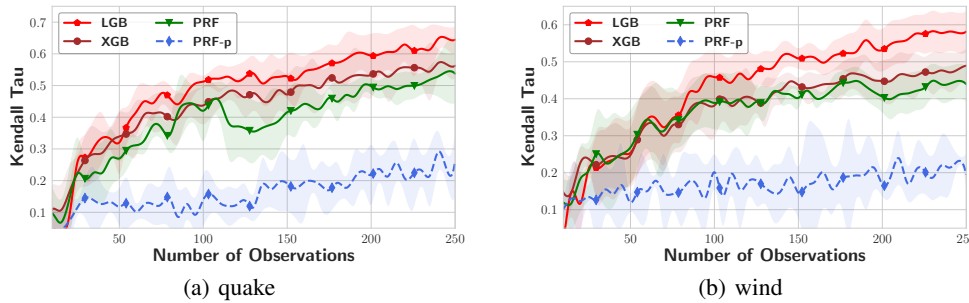

|     |     |
| --- | --- |
| (a) quake | (b) wind |

Figure 2: Kendall Tau correlation of different diversity surrogates with standard deviations shaded. '-p' refers to the performance surrogate while the other three are diversity surrogates.

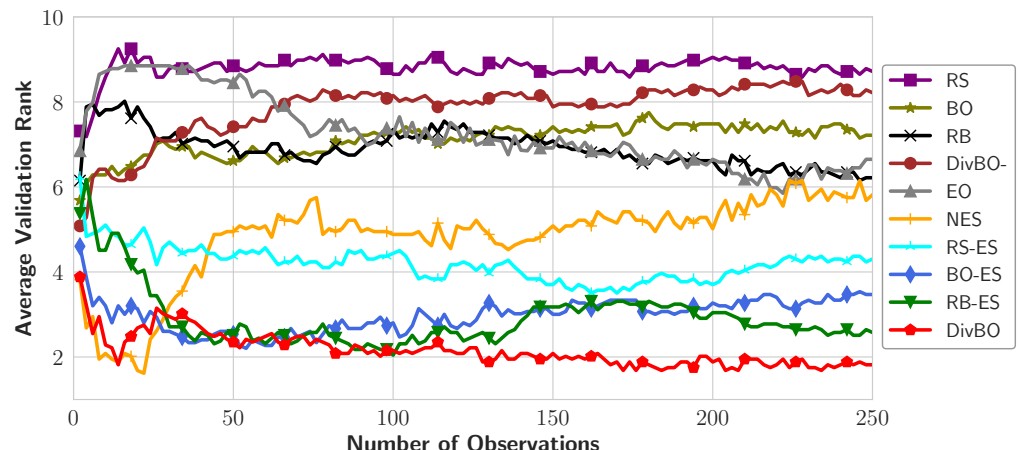

Figure 3: Average validation rank of 10 methods during optimization across 15 datasets. Note that ranks are a relative measure of performance (the rank of all methods add up to 55), and an improvement in validation error in one method may influence the rank of another one.

## 4.2 Evaluation of Diversity Surrogate

To show the soundness of DivBO, we first provide an analysis of the fitting capability of different diversity surrogates during optimization. We take LightGBM (LGB) [19], XGBoost (XGB) [8], and probabilistic random forest (PRF) as the candidates for diversity surrogate, where all three tree-based alternatives share the same time complexity $O(|D|^2 \log(|D|))$. We evaluate 300 randomly chosen configurations, among which up to 250 configurations are used to fit the surrogate, and the left 50 are used for surrogate evaluation. Figure 2 shows the Kendall Tau correlation between the surrogate predictive means across different runs and the ground-truth results on two different datasets.

We observe that, in general, the correlation of the diversity surrogate improves as the observations increase. Among the three alternatives, XGBoost performs slightly better than PRF, and LightGBM performs better than the other two alternatives. Remarkably, LightGBM achieves a strong Kendall Tau correlation of 0.65 and 0.58 on quake and wind when fitted with 250 observations, respectively. We also evaluate the performance surrogate (PRF-p) used in BO. Its correlation over the number of observations is much lower than the diversity surrogate, and the correlation is only 0.26 and 0.19 on quake and wind when fitted with 250 observations. The reason is that the search space for CASH problems is too large, and the performance surrogate can not be fitted well using limited observations $D$. While the number of pair-wise samples is much larger than non-pair-wise samples (i.e., $|D|^2$ vs. $|D|$), the diversity surrogate captures more information from the observations. As a result, the diversity surrogate can fit the diversity relationship between two configurations well and enjoys a

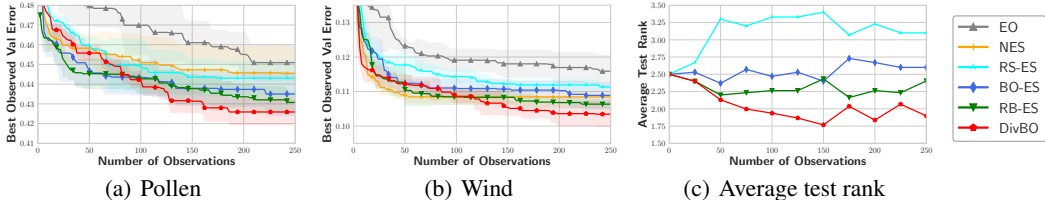

|            |            |            |
|:----------:|:----------:|:----------:|
| (a) Pollen | (b) Wind   | (c) Average test rank |

Figure 4: Figures (a) and (b): Best observed validation error with standard deviations shaded on two datasets. Figure (c): Average test rank of post-hoc ensemble designs across 15 datasets.

relatively strong correlation given a limited budget. In the following, we apply LightGBM as the diversity surrogate and evaluate DivBO on real-world AutoML tasks.

### 4.3 Evaluation of DivBO

**Performance Analysis.** In this section, we evaluate DivBO on 15 real-world CASH problems. Figure 3 shows the rank of validation error of all compared methods during optimization. We get four observations from the figure: 1) Ensemble learning indeed helps improve the performance of CASH results. Though Rising Bandit (RB) is competitive, the rank of non-ensemble methods (RS, BO, DivBO-) at the $250^{th}$ iteration is only 8.72, 7.21, and 8.22 among 10 compared methods, respectively; 2) Among methods with ensemble learning, the post-hoc designs outperform the other designs for ensemble learning (NES, EO). Note that, random search with ensemble selection (RS-ES) is a strong baseline. It achieves a rank of 4.30 at the $250^{th}$ iteration, which is better than that of NES (5.82) and EO (6.65). The reason is that the evolutionary algorithm in NES is not suitable for large search space while the performance of EO fluctuates if a poor learner is added into its fixed ensemble during optimization; 3) Among post-hoc designs, DivBO outperforms the other ones. While the second-best baseline achieves a rank of 2.58 at the $250^{th}$ iteration, the rank of DivBO is 1.82, and it consistently outperforms the other baselines after about 100 iterations. 4) We also find that DivBO- performs worse than BO without ensemble learning. The reason is that, the consideration of diversity is useless when searching for the single best classifier. But with the aid of the performance surrogate, DivBO- still performs better than random search. To demonstrate the validation error on specific datasets, we also plot the best-observed validation errors during optimization for six methods with ensemble learning in Figures 4(a) and 4(b). Similar to the trend of rank, DivBO exceeds the second-best baseline RB-ES after about 100 iterations. We further compare the number of required iterations for DivBO to achieve the same best validation errors as other methods. Concretely, DivBO achieves 1.52-1.54x and 1.67-2.53x speedups relative to RB-ES and BO-ES on the two datasets. We provide ablation study on weight scheduling and comparison with other intuitive designs in Appendix A.5.

Table 1 demonstrates the test errors and average rank on 15 datasets. We observe that the rank value of DivBO on test error is sometimes not consistent with that on validation error. The reason is that the distributions of the validation and test set are not exactly the same [18]. In other words, the configuration with the best validation error may not be the one with the best test error. But overall, post-hoc ensemble designs outperform other methods for ensemble learning, among which DivBO achieves the best test error on 10 out of 15 datasets, and its average rank is 1.73. The second best baseline is RB-ES, which performs the best on 3 out of 15 datasets, and its average rank is 2.90. The test rank of 4 post-hoc ensemble designs during optimization is also presented in Figure 4(c).

To check whether the improvement of DivBO is statistically significant, we conduct the Wilcoxon signed-rank test on each dataset given two methods. The difference is significant when the value $p \leq 0.05$. We count the number of datasets if 1) the mean error of DivBO is lower, and the difference is statistically significant (B); 2) the difference is not statistically significant (S); and 3) the mean error of DivBO is higher, and the difference is statistically significant (W). The results are presented in Table 2. Though RB-ES is a strong baseline, we observe that DivBO performs no worse than RB-ES on 12 datasets and better on 8 datasets. In Appendix A.5, we provide additional experiments to study the effects of different ensemble strategies and ensemble sizes.

Table 1: Test error (%) with standard deviations and the average rank across different datasets.

| Method | amazon_employee | bank32nh | cpu_act | cpu_small | eeg | elevators | house_8L | pol |
|---|---|---|---|---|---|---|---|---|
| | **CASH Methods** | | | | | | | |
| RS | $5.32 \pm 0.08$ | $18.43 \pm 0.79$ | $6.17 \pm 0.45$ | $7.57 \pm 0.41$ | $6.58 \pm 0.52$ | $10.22 \pm 0.68$ | $11.32 \pm 0.26$ | $1.65 \pm 0.12$ |
| BO | $5.26 \pm 0.06$ | $18.46 \pm 0.50$ | $5.69 \pm 0.38$ | $7.76 \pm 0.92$ | $5.47 \pm 1.35$ | $10.25 \pm 0.87$ | $11.20 \pm 0.19$ | $1.59 \pm 0.50$ |
| RB | $5.27 \pm 0.08$ | $18.33 \pm 0.32$ | $5.70 \pm 0.31$ | $7.53 \pm 0.35$ | $4.70 \pm 1.05$ | $9.77 \pm 0.20$ | $11.12 \pm 0.15$ | $1.39 \pm 0.03$ |
| | **Methods for Ensemble Learning** | | | | | | | |
| EO | $5.19 \pm 0.27$ | $18.47 \pm 0.53$ | $5.85 \pm 0.62$ | $7.42 \pm 0.45$ | $3.54 \pm 0.76$ | $10.34 \pm 0.61$ | $11.14 \pm 0.21$ | $1.43 \pm 0.13$ |
| NES | $5.32 \pm 0.38$ | $18.21 \pm 0.32$ | $5.59 \pm 0.20$ | $7.28 \pm 0.84$ | $\mathbf{2.68 \pm 0.73}$ | $9.75 \pm 0.30$ | $11.28 \pm 0.48$ | $1.74 \pm 0.35$ |
| | **Post-hoc Designs** | | | | | | | |
| RS-ES | $5.29 \pm 0.15$ | $18.30 \pm 0.66$ | $5.70 \pm 0.26$ | $7.21 \pm 0.30$ | $4.45 \pm 0.22$ | $9.51 \pm 0.28$ | $11.21 \pm 0.38$ | $1.39 \pm 0.15$ |
| BO-ES | $5.25 \pm 0.15$ | $18.41 \pm 0.39$ | $5.50 \pm 0.47$ | $7.16 \pm 0.28$ | $3.55 \pm 0.78$ | $9.61 \pm 0.36$ | $11.06 \pm 0.33$ | $1.35 \pm 0.18$ |
| RB-ES | $5.21 \pm 0.11$ | $\mathbf{18.07 \pm 0.57}$ | $5.58 \pm 0.20$ | $7.08 \pm 0.22$ | $2.86 \pm 0.92$ | $10.01 \pm 0.15$ | $10.81 \pm 0.27$ | $1.36 \pm 0.16$ |
| DivBO | $\mathbf{5.16 \pm 0.09}$ | $18.35 \pm 0.32$ | $\mathbf{5.37 \pm 0.23}$ | $\mathbf{7.04 \pm 0.29}$ | $3.26 \pm 0.84$ | $\mathbf{9.40 \pm 0.28}$ | $\mathbf{10.80 \pm 0.22}$ | $\mathbf{1.34 \pm 0.17}$ |

| Method | pollen | puma32H | quake | satimage | spambase | wind | 2dplanes | Average Rank |
|---|---|---|---|---|---|---|---|---|
| | **CASH Methods** | | | | | | | |
| RS | $51.53 \pm 0.53$ | $10.59 \pm 1.26$ | $48.17 \pm 1.88$ | $10.00 \pm 0.72$ | $7.14 \pm 1.23$ | $14.66 \pm 0.58$ | $7.21 \pm 0.08$ | 8.37 |
| BO | $49.64 \pm 3.39$ | $10.42 \pm 0.81$ | $46.88 \pm 2.16$ | $9.21 \pm 0.99$ | $6.45 \pm 0.84$ | $14.52 \pm 0.60$ | $7.15 \pm 0.06$ | 6.47 |
| RB | $49.79 \pm 1.15$ | $11.21 \pm 0.39$ | $47.98 \pm 1.56$ | $9.50 \pm 0.94$ | $6.71 \pm 1.05$ | $14.11 \pm 0.25$ | $7.20 \pm 0.05$ | 6.37 |
| | **Methods for Ensemble Learning** | | | | | | | |
| EO | $49.01 \pm 2.10$ | $9.74 \pm 1.60$ | $46.88 \pm 1.48$ | $9.42 \pm 1.11$ | $6.41 \pm 0.68$ | $14.62 \pm 0.48$ | $7.13 \pm 0.07$ | 5.77 |
| NES | $51.56 \pm 1.53$ | $10.63 \pm 0.54$ | $46.42 \pm 1.15$ | $8.66 \pm 0.95$ | $6.23 \pm 0.68$ | $14.25 \pm 0.50$ | $7.11 \pm 0.08$ | 5.00 |
| | **Post-hoc Designs** | | | | | | | |
| RS-ES | $49.69 \pm 1.63$ | $10.58 \pm 0.73$ | $46.79 \pm 1.57$ | $9.35 \pm 0.73$ | $6.45 \pm 0.23$ | $14.34 \pm 0.47$ | $7.11 \pm 0.12$ | 5.27 |
| BO-ES | $\mathbf{48.91 \pm 1.75}$ | $9.27 \pm 1.13$ | $46.10 \pm 2.52$ | $9.10 \pm 0.87$ | $6.38 \pm 0.64$ | $14.04 \pm 0.53$ | $7.07 \pm 0.08$ | 3.13 |
| RB-ES | $49.58 \pm 1.34$ | $\mathbf{7.85 \pm 0.43}$ | $46.70 \pm 1.34$ | $\mathbf{8.55 \pm 1.36}$ | $6.12 \pm 0.36$ | $13.98 \pm 0.45$ | $7.20 \pm 0.08$ | 2.90 |
| DivBO | $49.25 \pm 1.35$ | $8.07 \pm 0.99$ | $\mathbf{45.55 \pm 1.37}$ | $8.71 \pm 1.25$ | $\mathbf{5.91 \pm 0.45}$ | $\mathbf{13.93 \pm 0.42}$ | $\mathbf{7.00 \pm 0.08}$ | **1.73** |

Table 2: Counts of datasets when DivBO performs statistically better (B), the same (S), and worse (W) than compared three baselines.

(a) vs. RS-ES

| | B | S | W |
|---|---|---|---|
| DivBO | 13 | 2 | 0 |

(b) vs. BO-ES

| | B | S | W |
|---|---|---|---|
| DivBO | 12 | 1 | 2 |

(c) vs. RB-ES

| | B | S | W |
|---|---|---|---|
| DivBO | 8 | 4 | 3 |

**Diversity Analysis.** Finally, we analyze the optimization process of BO-ES, RB-ES, and DivBO. In Table 3, we show the validation errors of learners during optimization. Without ensemble, the single learner suggested by DivBO- performs worse than BO and RB. Note that, this does not mean that DivBO suggests bad configurations. We randomly evaluate 300 configurations from the search space. The mean result of those diverse configurations is better than 88% of the random configurations. To show how the diversity during the search process affects the ensemble, we use the update times of the temporary pool as a metric and the results are shown in Table 4. Since the temporary pool is built in the same way as the final ensemble, if the temporary pool changes, the configuration suggested at the previous iteration is included in the pool. Therefore, a change of the temporary pool at least indicates the suggested configuration affects the current ensemble. However, though the pool changes, the performance may not be improved due to the greedy mechanism, and thus we count the effective update times (i.e., the pool changes and the validation error of the ensemble decreases). As the pool updates very frequently in the beginning, we only calculate the mean effective update times of DivBO, RB-ES, and BO-ES on all datasets during the last 50 and 100 iterations. The pool is relatively stable in the last 50 iterations, which also indicates a budget of 250 iterations is sufficient for the datasets. We observe that, on average, DivBO will improve the temporary pool at least once in the last 50 iterations. While the difference between BO-ES and DivBO is the diversity part, we attribute this frequency gain to the use of diversity during the search process.

In Figure 5, given the total budget of 250 evaluation iterations, we plot the minimum diversity of the suggested learners (solid lines) after 200 iterations. The minimum diversity is defined as the diversity value (Equation 3) with the most similar learner in the ensemble built at the **previous** iteration. The minimum diversity of suggestions given by BO-ES and RB-ES is around 0.05, which indicates that the configuration suggestion is similar to one of the configurations in the previous ensemble. While for DivBO, the diversity is around 0.23, which is much higher than those of BO-ES and RB-ES.

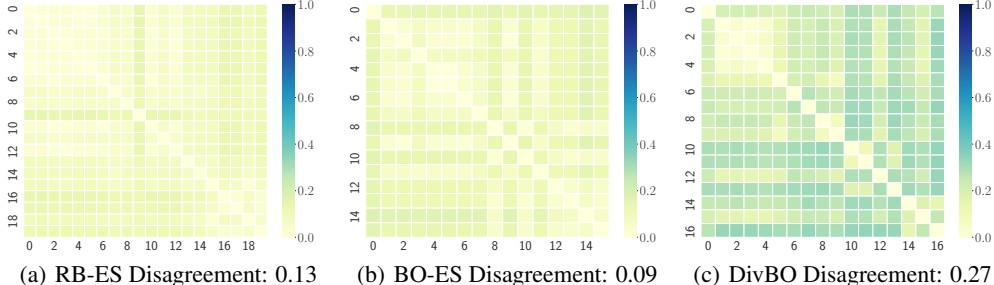

| (a) RB-ES Disagreement: 0.13 | (b) BO-ES Disagreement: 0.09 | (c) DivBO Disagreement: 0.27 |

Figure 6: Pair-wise diversity of learners in the post-hoc ensemble on quake. The learners in each method are numbered in order of observation.

Table 3: Val errors (%) of single learners during optimization.

|  | BO-ES | RB-ES | DivBO |
|---|---|---|---|
| Val Errors (%) | $44.03 \pm 2.58$ | $43.85 \pm 2.49$ | $44.33 \pm 2.67$ |

Table 4: Effective pool updates during optimization.

|  | BO-ES | RB-ES | DivBO |
|---|---|---|---|
| Counts (last 100) | 1.8 | 2.1 | 3.4 |
| Counts (last 50) | 0.6 | 0.8 | 1.5 |

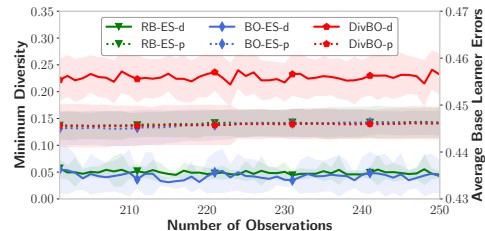

Figure 5: Minimum diversity of suggested learners (solid) and average error (dash) of base learners in the ensemble on quake.

In addition, we plot the average validation error of base learners in the current ensemble (dash lines) in Figure 5. The average error at the $250^{th}$ iteration given by the three post-hoc methods are quite similar (around $44.60\%$), which shows that applying the diversity surrogate in DivBO will not degenerate the pool of base learners. We further plot the pair-wise diversity between unique learners in the final ensemble in Figure 6 and present the average pair-wise predictive disagreement [38] in the caption. Generally, the larger disagreement is, the more diverse the learners in the ensemble are. Since ensemble selection selects base learners with replacement, the number of unique learners in the ensemble is different in each independent run. Though the average performance of base learners in the three post-hoc methods are similar, for RB-ES and BO-ES, the base learners are quite similar to each other. While for DivBO, we find that the learners are more diverse. Specifically, the learners that are found later (with a larger number) are more diverse with each other. The reason is that the diversity surrogate becomes more precise when fitted with more observations, and thus it can suggest more diverse configurations. This observation indicates that DivBO is able to generate a more diverse ensemble while ensuring the performance of base learners.

## 5   Conclusion

In this paper, we introduced DivBO, a diversity-aware framework based on Bayesian optimization to solve CASH problems for ensemble learning. In DivBO, we proposed to use a diversity surrogate to model the relationship between two configurations, and combined the ranking values of the performance and diversity surrogates with a saturating weight. Through empirical study, we showed that the prediction of the diversity surrogate achieves a satisfactory correlation with ground-truth results, and DivBO outperforms post-hoc designs in recent AutoML systems and other baselines for ensemble learning in CASH problems.

## Acknowledgments

This work is supported by NSFC (No. 61832001) and ZTE-PKU Joint Laboratory for Foundation Software. Yang Li and Bin Cui are the corresponding authors.

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
