# OpenReview forum: "DivBO: Diversity-aware CASH for Ensemble Learning"
_NeurIPS.cc/2022/Conference — NeurIPS 2022 Accept_

### Official Review · Reviewer_anrp · 2022-07-03

**Rating:** 7
**Confidence:** 4
**Soundness:** 3 good
**Presentation:** 4 excellent
**Contribution:** 3 good

**Summary:**

This work proposed a method to improve the ensemble performance under the CASH framework. The basis of the method is based on observations by many previous work that a diverse set of base learners will usually improve ensemble performance. The method jointly searches for configurations not only with a good objective but also diverse ones compared to the configurations that have been evaluated.

To achieve that, the paper proposed a surrogate to measure diversity between a pair of configurations and combine the acquisitions for quality and diversity through a weighted ranking. Experiments demonstrated improved performance compared to well thoughted baselines.


**Questions:**

- I am not sure about “... while ensuring the performance of base learners” in line 328. Is the average performance a good measure to claim that? As we have seen in Figure 3 and the author also wrote in line 286, “We also find that DivBO- performs worse than BO without ensemble learning”. So the best base learners found by DivBO are clearly worse than others.
- Minor: In Figure 5, “-d” and “-p” are not explained.


**Limitations:**

- I would suggest the authors highlight the potential overfitting issue, which has been observed in their experiments.
- Also, the ensemble can lead to slow inference and one should still consider a single model if inference time is critical.


**Strengths And Weaknesses:**

The strengths of the paper include:
- A novel idea to consider both diversity and performance during CASH.
- Very good presentation quality with clear structure and detail.
- Informative experiments result to support the claims.

The weakness of the paper include:
- Since this is an empirical paper, I would expect evaluations on more datasets, especially given all the datasets are less than 20k rows. For example the [OpenML benchmark suites](https://openreview.net/forum?id=OCrD8ycKjG).

Nice to have:
- There is a strong baseline called [AutoGluon Tabular](https://github.com/awslabs/autogluon), which reports competitive empirical results by training with default HPs and then doing multi-layer stacking. It would be nice to know how the proposed method compares to it.

==== After reading the author's response ===

The authors addressed all my questions and comments, I will increase the score to 7.

---

> ### Author Response · Authors · 2022-08-02
> **Response to Reviewer anrp, Part 1**
>
> # Response to Reviewer anrp
> Thanks for your constructive feedback! We believe that addressing this feedback will make our paper significantly stronger. The detailed response to each question is as follows,
>
> ### Q1. Results on more datasets.
> As suggested, we add two more datasets from OpenML which contain more than 20k instances (40768 for 2dplanes and 32769 for amazon_employee). We provide the test errors (%) of competitive post-ensemble methods as follows,
>
> |  | RS-ES | BO-ES | RB-ES| DivBO|
> | - | :-: | :-: | :-: | :-: |
> | 2dplanes    |  7.11±0.12  |   7.07±0.08  |  7.20±0.08  |  **7.00±0.08**  |
> | amazon_employee |  5.29±0.15  | 5.25±0.15 | 5.21±0.11 |  **5.16±0.09** |
>
>
> While the results on amazon_employee are not significant, we apply the Wilcoxon signed-rank test and the p-value is 0.04 (< the threshold 0.05), which means the improvement of DivBO over RB-ES is statistically significant. We will definitely add them to our experiments in the final manuscript.
>
> ### Q2. Auto-Gluon Tabular as a new baseline.
> The search space plays a significant role in CASH optimization. As DivBO is an algorithm framework rather than a system, we compare it with other baselines by using the same search space (i.e., auto-sklearn space). However, AutoGluon applies a more compact space than auto-sklearn, and it's not fair to directly compare DivBO on auto-sklearn search space with AutoGluon. We agree with the reviewer that Auto-Gluon Tabular is a strong baseline, and should be compared in the paper. To make a relatively fair comparison, we reproduce a similar search space of AutoGluon except for the specified neural networks due to implementation difficulty. The results on five datasets are as follows,
>
> | Test Errors (%)  | elevators | house_8L | pol | quake | wind |
> | - | :-: | :-: | :-: | :-:| :-: |
> | AutoGluon Tabular    |  9.10±0.00  |  **9.98±0.00**  | 1.23±0.00 | 44.72±0.00 | 14.37±0.00 |
> | DivBO (AutoGluon space)      |  **9.01±0.11**  |  10.06±0.17  | **1.18±0.07** | 44.75±0.60 | **14.24±0.18** |
>
>
> Note that, the search space affects the results a lot. For example, AutoGluon's results on wind are worse than RS-ES using the auto-sklearn space. However, AutoGluon's results on the other four datasets are better than most of the results using the auto-sklearn space, which is consistent with the observation that AutoGluon often outperforms auto-sklearn. The reason may be that AutoGluon is equipped with a well-designed search space, which kicks out less reliable algorithms on modern datasets (e.g., Naive Bayes) and adds strong ones (e.g., Catboost). Note that, the variance of AutoGluon's results is zero because it fixes the random seed in its inner design. In addition, we observe an error decrease when using DivBO in this search space. Concretely, the improvement is **statistically** significant on three datasets, not significant on one (quake), and slightly worse on the other one (house_8L).
>
> ### Q3. Explanation of ensemble diagnostics.
> The sentence “We also find that DivBO- performs worse than BO without ensemble learning” means that the best learner found by DivBO- is worse than BO. This is not contradictory to the results that DivBO performs better than BO-ES. The performance of an ensemble does not depend on the best observed learner alone, but on all the learners in the ensemble. The average learner performance is a metric that measures the overall strength of base learners in the ensemble, which is also used in previous ensemble learning work \[1\] as one of the ensemble diagnostics.
>
> We also apply the predictive disagreement \[2\] to analyze the diversity of the ensemble. Given two learners, the pair-wise predictive disagreement computes the ratio of disagreed instances for all instances. (Disagreement happens when a learner classifies correctly but the other does not.) The disagreement for an ensemble is computed by averaging the pair-wise disagreement given all pairs of learners in the ensemble. Generally, the larger disagreement is, the more diverse the learners in the ensemble are. We compute the disagreement at the 250-th iteration in Figure 5, and the mean results are as follows,
>
> |   | BO-ES | RB-ES | DivBO |
> | - | :-: | :-: | :-: |
> | Disagreement     |  0.09  |  0.13  | 0.27 |
>
> While the average performance of base learners is similar, the ensemble built by DivBO enjoys a higher disagreement value (diversity), thus achieving better test accuracy.
>
> [1] S. Zaidi, A. Zela, T. Elsken, C. C. Holmes, F. Hutter, and Y. Teh. Neural ensemble search for uncertainty estimation and dataset shift. Advances in Neural Information Processing Systems, 34, 2021.
>
> [2] Tang, E. Ke, Ponnuthurai N. Suganthan, and Xin Yao. "An analysis of diversity measures." Machine learning 65.1 (2006): 247-271.

---

> > ### Author Response · Authors · 2022-08-02
> > **Response to Reviewer anrp, Part 2**
> >
> > ### Q4. Explanation of captions.
> > ‘-d’ means the correlation of the diversity surrogates with ground-truth diversity, while ‘-p’ means the correlation of the performance surrogate with ground-truth performance. We will add more explanations.
> >
> >
> > ### L1. Potential overfitting issue.
> > As suggested, we will highlight the issue in the limitation (see L2 below).
> >
> > ### L2. Limitation.
> > To clarify the limitation, we will add a paragraph in Section 4.4.  The updated limitations will be as follows,
> >
> > Limitation. a) The use of ensemble leads to higher inference latency than using the single best learner (approximately K times where K is the number of learners in the ensemble). This latency can be reduced with the aid of parallel computing if we have sufficient computational resources. In addition, as ensemble selection is fitted on the validation set, there's a risk of overfitting on the test set for small datasets. b) DivBO using Equation 3 as the diversity function can not directly support algorithms that only predict class labels (e.g., SVC). Though DivBO still works by converting the predicted labels to class probability (like [1, 0, …]), other diversity functions can be developed to support those algorithms better.

---

> > > ### Comment · Reviewer_anrp · 2022-08-04
> > > **Nice response**
> > >
> > > Thank you for the detailed response, I will increase my score.

---

### Official Review · Reviewer_15EU · 2022-07-07

**Rating:** 6
**Confidence:** 5
**Soundness:** 3 good
**Presentation:** 3 good
**Contribution:** 3 good

**Summary:**

In this work, the authors aim to improve CASH solutions by favoring diverse predictors during the optimization which are more appropriate during post-hoc ensembling. In order to achieve this, the authors adapt the acquisition function in the Bayesian optimization by a diversity term. Therefore, selected candidates which are a trade-off between predictive performance and a diverse addition to the current ensemble. The authors compare their method against standard CASH methods, automated ensemble learning methods and AutoML methods + posthoc ensembling on 15 datasets.

**Questions:**

Is the improvement statistically significant?

How important are the most diverse models in the final ensemble?

**Limitations:**

Discussion on problems of ensemble models could be added.

Maybe the authors can be more precise about where they describe their limitations. The section referred to in the checklist does not seem to discuss limitations.

**Strengths And Weaknesses:**

**Summary**

The idea is reasonable, is of academic relevance and to some degree relevant for practitioners. The method is clearly described. The empirical evaluation raises some concerns regarding the usefulness of the proposed method. While the method usually ranks best, the improvement over a random search with post-hoc ensembling does not seem to be statistical significant and no significance test is provided to show otherwise. Further ablation studies could have been conducted to provided deeper understanding about the importance of diverse models and the hyperparameter sensitivity.

**Details**

It is common practice to select upfront a diverse set of algorithms, train a couple of them with different hyperparameters and use a weighted average across them. This raises the question how the proposed method compares against this simple baseline. A method in the literature that makes use of this fact is [1]: They decompose the CASH problem in K HPO problems, solve those and then use post-hoc ensembling. This will also result in diverse, well-performing models, mimics the human strategy but does not explicitly consider the search for diverse candidates given an ensemble. It would be interesting to understand whether this is important and whether the proposed method actually makes efficient use of it. At the moment we only know that the considered baseline methods are skewed towards better performing algorithms which reduces diversity. The solution in the paper mentioned above seems to be a very simple solution to overcome this problem.

Figure 6 is great since it demonstrates that DivBO is indeed finding ensembles with higher diversity. However, this is not particularly surprising given that the pool of candidates for the post-hoc ensembling is more diverse from the beginning. I think it would be more interesting to understand the importance of the diverse models for the ensemble. This could be done by using a greedy post-hoc ensembling approach and report the order of models or report the performance of the ensembling when removing the most diverse models. Optimally, those results are reported across all datasets and not a single one.

While the test error in Table 3 of DivBO is oftentimes the lowest, the improvement over BO-ES or even RS-ES is small and given the high standard deviation, I expect the improvement to not be statistically significant. It would be great if the authors could provide a significance test.

Addition of two new hyperparameters. The provided sensitivity analysis on a single dataset is insufficient to judge how drastically the optimal setting changes with different datasets. The authors mention that the sensitivity analysis was the motivation for the final choice of the hyperparameter settings. However, the optimal solution is at the edge of the grid. How does the grid look like for larger or smaller taus and larger betas? Getting an idea about the sensitivity based on only one dataset is hard. It would be important to understand whether the best settings are significantly different across datasets.


The use of an ensemble in the first place is a limitation which is not discussed. In practice, ensembles may be well-performing but they are typically not used since they are very hard to understand, have higher inference time and are harder to maintain.

**References**

[1] Martin Wistuba, Nicolas Schilling, Lars Schmidt-Thieme: Automatic Frankensteining: Creating Complex Ensembles Autonomously. SDM 2017: 741-749

---

> ### Author Response · Authors · 2022-08-02
> **Response to Reviewer 15EU, Part 1**
>
> Thanks for your constructive feedback! We believe that addressing this feedback will make our paper significantly stronger. The detailed response to each question is as follows,
>
> ### Q1. A new baseline.
> Thanks for the suggestion. We evaluate the simplest version as suggested, which tunes each algorithm for the same budget and then builds a post-hoc ensemble. In fact, it is a simplified version of the baseline RB-ES, in which RB-ES eliminates some of the algorithms after several iterations. We name it kBO-ES and present the results as follows,
>
> | Test Errors (%)  | elevators | house_8L | pol | quake | wind |
> | - | :-: | :-: | :-: | :-:| :-: |
> | RS-ES     |  9.51±0.28  |  11.21±0.38  | 1.39±0.15 | 46.79±1.57 | 14.34±0.47 |
> | kBO-ES | 9.55±0.32 | 11.18±0.34 | 1.39±0.16 | 46.81±1.48 | 14.29±0.45 |
> | DivBO      |  **9.40±0.28**  |  **10.80±0.22**  | **1.34±0.17** | **45.55±1.37** | **13.93±0.42** |
>
> We observe that the results of kBO-ES are quite similar to RS-ES (Random search with ensemble selection). The reason is that the search space contains a lot of algorithms while the budget is quite limited (250 iterations). Each algorithm can only be tuned about 22 times. For each algorithm, we also need to tune the feature engineering operators (>50 HPs in auto-sklearn search space), and thus the BO surrogate for each algorithm is under-fitted. Therefore, Bayesian optimization for each algorithm performs like random search. kBO-ES is an intuitive method but seems to perform not competitively when the search space is very large.
>
> ### Q2. Experiments on diversity.
> As suggested by the reviewer, we add the experiments on the influence of removing the most diverse learners from the final ensemble. While DivBO is extended from naive BO, we evaluate the mean influence of removing the top-3 diverse models from the final ensemble. As the ensemble is built on the validation set, we present the validation error gaps on five datasets as follows, (a positive gap means removing the models leads to an error increase)
>
> | Gap (%)  | elevators | house_8L | pol | quake | wind |
> | - | :-: | :-: | :-: | :-:| :-: |
> | BO-ES |  +0.21  |  -0.04  | +0.02 |  +0.98  | +0.38 |
> | DivBO |  +0.38  |  +0.15  | +0.01 |  +1.28  | +0.61 |
>
> We observe that generally, removing the most diverse learners from DivBO leads to a larger error increase than BO-ES. As it's easy to learn good learners on the dataset pol (the accuracy of almost all the learners in the ensemble is above 98%), removing learners affects quite little on the ensemble performance.
>
> Besides the experiments suggested by the reviewer, we also add an overall analysis of the final ensemble using ensemble diagnostics. We apply the predictive disagreement[1] to further analyze the diversity of the ensemble. Given two learners, the pair-wise predictive disagreement computes the ratio of disagreed instances for all instances. (Disagreement happens when a learner classifies correctly but the other does not.) The disagreement for an ensemble is computed by averaging the pair-wise disagreement given all pairs of learners in the ensemble. Generally, the larger disagreement is, the more diverse the learners in the ensemble are. We compute the disagreement at the 250-th iteration in Figure 5, and the mean results are as follows,
>
> |   | BO-ES | RB-ES | DivBO |
> | - | :-: | :-: | :-: |
> | Disagreement     |  0.09  |  0.13  | 0.27 |
>
> While the average performance of base learners is similar, the final ensemble built by DivBO enjoys a higher disagreement value (diversity), thus achieving better test accuracy.

---

> > ### Author Response · Authors · 2022-08-02
> > **Response to Reviewer 15EU, Part 2**
> >
> > ### Q3. Significance test.
> > To check whether the improvement of DivBO is statistically significant, we conduct the Wilcoxon signed-rank test on each dataset given two methods. The difference is significant when the value $p\leq0.05$ \[1\]. We count the number of datasets if 1) DivBO is better than the other method, and the difference is statistically significant (**B**); 2) the difference is not statistically significant (**S**); and 3) DivBO is worse, and the difference is statistically significant (**W**). For each dataset, the rank of DivBO is 1 for **B**, 1.5 for **S**, and 2 for **W**. We compute the pair-wise rank by averaging the rank on 15 datasets. The results are presented as follows,
> >
> > **DivBO vs. RB-ES.** The pair-wise ranks of DivBO and RB-ES on all datasets are 1.33 and 1.67, respectively. We agree that RB-ES is a strong baseline. Through the significance test, we observe that DivBO performs no worse than RB-ES on 12 datasets and better on 8 datasets.
> >
> >
> > |   | B | S | W |
> > | - | :-: | :-: | :-: |
> > | DivBO     |  8  |  4  |  3  |
> >
> > **DivBO vs. BO-ES.** The pair-wise ranks of DivBO and BO-ES on all datasets are 1.17 and 1.83, respectively. While the DivBO framework is extended from BO, DivBO generally performs better than BO. Concretely, DivBO performs no worse than BO-ES on 13 datasets and better on 12 datasets.
> >
> >
> > |   | B | S | W |
> > | - | :-: | :-: | :-: |
> > | DivBO     |  12  |  1  | 2 |
> >
> > **DivBO vs. RS-ES.** The pair-wise ranks of DivBO and RS-ES on all datasets are 1.07 and 1.93, respectively. DivBO performs no worse than RS-ES on all datasets.
> >
> >
> > |   | B | S | W |
> > | - | :-: | :-: | :-: |
> > | DivBO     |  13  |  2  | 0 |
> >
> > \[1\] Wilcoxon, Frank. Individual comparisons by ranking methods. Breakthroughs in statistics. Springer, New York, NY, 1992. 196-202.
> >
> > ### Q4. Sensitivity analysis.
> > As suggested by the reviewer, we extend the 4\*4 grid to the 4\*5 grid for sensitivity analysis and add another dataset. The results are as follows,
> >
> > | Spambase | $\tau$=0.05 | $\tau$=0.1 | $\tau$=0.2 | $\tau$=0.4 | $\tau$=0.8 |
> > | - | :-: | :-: | :-: | :-: | :-:|
> > | $\beta$=0.025  |  96.59  |  96.30  | 96.20 |  *96.67*  | 96.34 |
> > | $\beta$=0.05 |  96.12  |  *96.59*  |  **96.78**  |  *96.74*  | 96.41 |
> > | $\beta$=0.1 | 96.01  | 95.98 |  96.27  |  96.23  | 96.30 |
> > | $\beta$=0.2 | 95.76 | 95.80 |  95.80  |  95.76 | 95.68 |
> >
> >
> > | House_8L | $\tau$=0.05 | $\tau$=0.1 | $\tau$=0.2 | $\tau$=0.4 | $\tau$=0.8 |
> > | - | :-: | :-: | :-: | :-: | :-:|
> > | $\beta$=0.025  |  89.60  |  *89.98*  | *89.99* |  89.65  | 90.13 |
> > | $\beta$=0.05 |  89.66  |  89.96  |  *90.10*  |  **90.40**  | *89.95* |
> > | $\beta$=0.1 | 89.96  | 89.76 |  89.25  |  89.83  | *89.98* |
> > | $\beta$=0.2 | 89.37 | 89.40 |  89.60  |  89.34 | 89.17 |
> >
> > Remind that $\beta$ is the maximum of diversity importance and $\tau$ controls the speed of approaching saturation. We observe that a large $\beta$ (0.2) leads to a clear accuracy drop, and we suggest using a $\beta=0.05$. However, we need to tune $\tau$ to achieve the best results on different datasets. The reason may be that the difficulty for different datasets to find good configurations are different. As DivBO builds on the intuition that we need to focus on accuracy rather than diversity in early iterations, a smaller $\tau$ is required if it's difficult to find accurate learners in early iterations. The suggested region for tuning $\tau$ is \[0.1,0.8\]. In our paper, we use 0.2 by default, but a tuned $\tau$ may achieve better results. As suggested, we will update the sensitivity analysis and add the analysis in Section 3.2.
> >
> > ### Q5. Limitation.
> > To clarify the limitation, we will add a paragraph in Section 4.4. The updated limitations will be as follows,
> >
> > Limitation. a) The use of ensemble leads to higher inference latency than using the single best learner (approximately K times where K is the number of learners in the ensemble). This latency can be reduced with the aid of parallel computing if we have sufficient computational resources. In addition, as ensemble selection is fitted on the validation set, there's a risk of overfitting on the test set for small datasets. b) DivBO using Equation 3 as the diversity function can not directly support algorithms that only predict class labels (e.g., SVC). Though DivBO still works by converting the predicted labels to class probability (like [1, 0, …]), other diversity functions can be developed to support those algorithms better.

---

> > > ### Comment · Reviewer_15EU · 2022-08-03
> > > **Thank you**
> > >
> > > Thank you for addressing all my comments. I will update my score accordingly.

---

### Official Review · Reviewer_t73G · 2022-07-11

**Rating:** 6
**Confidence:** 4
**Soundness:** 3 good
**Presentation:** 3 good
**Contribution:** 2 fair

**Summary:**

This paper focuses on the problem of CASH (Combined Algorithm Selection and Hyperparameter Optimization) to automatically configure machine learning models (or pipelines), and seeks to improve the quality of the final ensemble generated by various AutoML solutions using the model/pipelines configurations tried during the optimization. Based on the intuition that diversity among the base models improves the quality of any ensemble, this paper seeks to modify a CASH solver to generate a more diverse set of base models, while still maintaining the predictive performance of the base models. To this end, the paper presents a notion of diversity between any two pair of model configurations, and shows how it can be incorporated into a Bayesian Optimization framework (BO) by proposing a new acquisition function that combines the predictive performance and diversity in a way that the predictive performance of selected models is maintained while improving the diversity among the base models. Empirically, the paper demonstrates the ability of the proposed DivBO scheme to generate diverse set of base models (based on their proposed definition of diversity) while solving the CASH problem. The empirical results also show that ensembles created by DivBO improve upon ensembles created by diversity agnostic CASH solvers.



**Questions:**

- While DivBO- is considered in the experiments, are there results (empirical or theoretical) highlighting the difference between DivBO and DivBO without the weight schedule in equation (5); that is, the weight is fixed to $\beta$ ($w = \beta$)? While it seems intuitive that DivBO with the weight schedule would perform better, it would be good to have empirical evidence.

- The paper spends a lot of space on how the surrogate model for the diversity metric is created and how it generalizes. This is useful detail but lot of it appears to be straightforward given observations of the metric much like most surrogate modeling. It would great to understand what were the challenges in the surrogate modeling of the diversity which motivated the authors to describe and study the surrogate modeling of the diversity metric in such detail, instead of studying, for example, the effect of the different choices in DivBO (such as effect of $\beta$ and $\tau$ in the weight schedule in the main paper, or the number of samples for the combined acquisition function optimization).

- The definition of the diversity acquisition function is not entirely clear. It appears from the definition and the following text that, in one trial, for each of the $\theta \in \mathcal{P}$, we sample from $M_{div}(\theta, x)$ and then take a minimum over all $\theta \in \mathcal{P}$, and take an average over $N$ such trials. Can you please explain why we are using a mean over minimum-of-per-theta samples and not just minimum of per-theta-sample-means? This choice seems a bit unintuitive and I would like to understand if this plays a significant role in DivBO, and if so, why?


- Is BO exactly the same as DivBO- where $w=0$ throughout the optimization (or equivalently BO-ES and DivBO)? Otherwise different performance surrogate function and/or acquisition function maximization strategies will lead to differences and it won't be clear if the difference between BO and DivBO is because of the diversity incorporation or just because of other differences in the BO configurations.

- In the evaluation in section 4.2, the correlation between the true diversity and predicted diversity with $M_{div}$ appears positive but still is only around 0.5 even with 250 configurations implying  $250^2$ observations to fit the $M_{div}$ surrogate model, which seems to be a large number of observations (albeit non-i.i.d. ones). Is there an explanation for why the $M_{div}$ quality is not much higher? What are the potential challenges?

- In the definition of diversity in equation (3), the predicted class probability $T_{x_i}(s)$ for some configuration $x_i$ and validation sample $s$ require that the trained model generates probabilities. How are the class probabilities generated from discriminative models such as SVC (linear or kernel) which do not inherently produce class probabilities but rather directly output class labels?


- In Figure 6, why do the different schemes have different number of base learners? It is weird to compare different baselines with different number of ensembles.


**Limitations:**

The authors claim in the Checklist that they discuss the limitations of the proposed scheme in Figure 3 and Section 4.3, it is not clear from the figure or the subsection what limitations of DivBO they have discussed.


**Strengths And Weaknesses:**

## Strengths

- The need for diversity in ensembles have been considered for a long time. It is great to see a CASH solver like DivBO that directly incorporates the desired diversity in the final ensemble within the CASH optimization instead of keeping the optimization and the ensembling completely separate as with most existing AutoML solutions that use CASH solvers. This is strong novel contribution.

- Given that diversity and predictive performance is not always aligned, the authors do a great job at clearly highlighting the challenges of incorporating diversity in a CASH solution and the intuitions guiding their proposed scheme. It is true that (i) there is no need to have high diversity from base models that do not perform well (and hence wont probably be part of the final ensemble), and (ii) high diversity should not be at the cost of predictive performance, since that form of diversity will not be useful for the final ensemble. To this end, the authors define an acquisition function for diversity that only relies on diversity from a pool of potential ensemble members, and is able to be incorporated with the acquisition function for the predictive performance with use of ranks instead of raw values.

- Another original strong idea in this paper is the way the predictive performance rank and the diversity rank are weighed in the final acquisition function (equation (5)). The use of a weight schedule is well motivated by the authors with the intuition that initially we want the CASH solver to focus on predictive performance (which allows us to get a strong pool of potential ensemble base learners), and gradually push the optimization to increase diversity without hurting the predictive performance too much.


- The authors provide a very concise but appropriate literature review.


## Weaknesses

- While it has been known that diversity in the base models of the ensemble can improve generalization, one key component is the definition of diversity. There is not much discussion on why the definition of diversity in equation (3) leads to improved ensemble performance (beyond the presented empirical results). As a reader, I would like a better motivation and understanding for this diversity metric since it is a critical part of this paper.

- Another point that is not clear is that, given that we are performing ensemble selection using the greedy selection scheme of Caruana et al (2004), would diversity play any role at all in the final ensemble selection? Given that we greedily select the next ensemble member which most improves the validation loss, what are conditions under which the increased diversity in base learners would lead to selection of the more diverse base learner compared to less diverse ones?

- From the results in Figure 3, it is not clear that DivBO is significantly better than RB-ES and BO-ES. Firstly, with so many baselines, it is more useful to consider pairwise comparisons (as seen in citation [24] and [A]) especially between closely performing baselines. Secondly, comparing the average rank without confidence intervals (or better, some form of Wilcoxon signed rank test of significance), it is not clear if DivBO improves over BO-ES and other baselines significantly because of the incorporated diversity. In fact, in Figure 4 and Table 3, the average performance improvement of DivBO against baselines is well within the predictive performance confidence intervals posted, undermining the expected improved generalization ability of an ensemble with diverse base models.



[A] Rakotoarison, Herilalaina, Marc Schoenauer, and Michèle Sebag. "Automated Machine Learning with Monte-Carlo Tree Search." IJCAI-19-28th International Joint Conference on Artificial Intelligence. International Joint Conferences on Artificial Intelligence Organization, 2019.


- Figure 5 (and also Figure 1) shows how different the next suggested configuration is to the ones in the current pool $\mathcal{P}$. Hence this highlights the ability of the combined acquisition function and its maximization (probably after the weight saturation) to propose diverse candidate configurations. However, it is not clear (i) how good is the predictive performance of these diverse configuration, (ii) how this improves the predictive performance of the ensemble. The authors also present the average validation performance of the base models in the ensemble pool in Figure 5, highlighting that the base models' performances do not drop because of the increased diversity, but it does not say anything about improving the ensemble performance -- it is quite possible that DivBO proposes diverse configurations but they are not included in the ensemble, in which case, neither the ensemble performance nor the average base model performance would be affected by the diversity. This figure does not seem to make it clear that the increased diversity is playing a significant role in improving the ensemble quality.

---

> ### Author Response · Authors · 2022-08-02
> **Response to Reviewer t73G, Part 1**
>
> Thanks for your constructive feedback! We believe that addressing this feedback will make our paper significantly stronger. The detailed response to each question is as follows,
>
> ### W1. A brief discussion on diversity definition.
> We will add a brief discussion on the definition of diversity in Section 3.1 as follows,
>
> The existing diversity measures can be generally divided into pair-wise and nonpair-wise measures. The nonpair-wise diversity [1] directly measures a set of learners in the ensemble. While the ensemble may change during each iteration, it’s difficult to model the diversity of an ensemble with a candidate configuration and multiple learners. Therefore, we use the pair-wise measures to simply learn the diversity of two given configurations. To this end, we follow the definition in previous research [2] that explicitly improves the diversity of neural networks and also shows satisfactory empirical results.
>
> [1] Z.-H. Zhou, Ensemble Methods: Foundations and Algorithms. Boca Raton, FL, USA: CRC, 2012.
>
> [2] W. Zhang, J. Jiang, Y. Shao, and B. Cui. Efficient diversity-driven ensemble for deep neural networks. In 2020 IEEE 36th International Conference on Data Engineering (ICDE), pages 73–84. IEEE, 2020.
>
> ### W2. The effect of diversity in ensemble selection.
> We agree that it's quite difficult to explain whether ensemble selection prefers high or low diversity during each selection round. But considering an extreme case where all the learners are exactly the same (with no diversity), there’s no gain when using ensemble selection. This at least indicates that building a good ensemble using ensemble selection still requires diversity. To show that suggesting diverse configurations during the search process improves the final ensemble, we add additional experiments. Concretely, 1) we **analyze the update frequency of the temporary pool** to show whether suggesting diverse configurations will improve the current ensemble, and 2) we **analyze the final ensemble based on another diversity metric 'disagreement'**. Please refer to 'W4. Diversity analysis' for setups, results, and analysis.
>
> ### W3. Significance test.
> To check whether the improvement of DivBO is statistically significant, as suggested by the reviewer, we conduct the Wilcoxon signed-rank test on each dataset given a pair of methods. The difference is significant when the value $p\leq0.05$ \[1\]. We count the number of datasets if 1) DivBO is better than the other method, and the difference is statistically significant (**B**); 2) the difference is not statistically significant (**S**); and 3) DivBO is worse, and the difference is statistically significant (**W**). For each dataset, the rank of DivBO is 1 for **B**, 1.5 for **S**, and 2 for **W**. We compute the pair-wise rank by averaging the rank on 15 datasets. The results are presented as follows,
>
> **DivBO vs. RB-ES.** The pair-wise ranks of DivBO and RB-ES on all datasets are 1.33 and 1.67, respectively. We agree that RB-ES is a strong baseline. Through the significance test, we observe that DivBO performs no worse than RB-ES on 12 datasets and better on 8 datasets.
>
>
> |   | B | S | W |
> | - | :-: | :-: | :-: |
> | DivBO     |  8  |  4  |  3  |
>
> **DivBO vs. BO-ES.** The pair-wise ranks of DivBO and BO-ES on all datasets are 1.17 and 1.83, respectively. While the DivBO framework is extended from BO, DivBO generally performs better than BO. Concretely, DivBO performs no worse than BO-ES on 13 datasets and better on 12 datasets.
>
>
> |   | B | S | W |
> | - | :-: | :-: | :-: |
> | DivBO     |  12  |  1  | 2 |
>
>
> **DivBO vs. RS-ES.** The pair-wise ranks of DivBO and RS-ES on all datasets are 1.07 and 1.93, respectively. DivBO performs no worse than RS-ES on all datasets.
>
>
> |   | B | S | W |
> | - | :-: | :-: | :-: |
> | DivBO     |  13  |  2  | 0 |
>
>
> \[1\] Wilcoxon, Frank. Individual comparisons by ranking methods. Breakthroughs in statistics. Springer, New York, NY, 1992. 196-202.

---

> > ### Author Response · Authors · 2022-08-02
> > **Response to Reviewer t73G, Part 2**
> >
> > ### W4. Diversity analysis.
> >
> > #### Predictive performance of suggested configurations
> > As suggested by the reviewer, we first provide the validation errors of configurations (without ensemble) suggested by BO-ES, RB-ES, and DivBO during the last 50 iterations on quake (the same settings as shown in Figure 5). Note that, here we present the errors but not the best observed errors. The variance is **very large** due to the high randomness of Bayesian optimization when balancing exploration and exploitation. The results are as follows,
> >
> > |   | BO-ES | RB-ES | DivBO |
> > | - | :-: | :-: | :-: |
> > | Val Errors (%)     |  44.03±2.58  |  43.85±2.49  | 44.33±2.67 |
> >
> > The results are consistent with Figure 3 that, without ensemble, the single learner suggested by DivBO- performs worse than BO and RB. Note that, this does not mean that DivBO suggests bad configurations. We randomly evaluate 300 configurations from the search space. The mean result of those diverse configurations is better than 88% of the random configurations. Below, we analyze the effects of suggesting those diverse configurations on the ensemble.
> >
> > #### Effective update frequency of ensemble
> > As mentioned by the reviewer, a very large proportion of configurations suggested during the search process will not affect the final ensemble. It's difficult to directly build the relationship between each suggested configuration and the final ensemble. To show how the diversity during the search process affects the ensemble, we use the update times of the temporary pool as a metric.
> >
> > As mentioned in the paper, the temporary pool is built in the same way as the final ensemble. In other words, the temporary pool is the final ensemble if the optimization stops at the previous iteration. If the temporary pool changes, the configuration suggested at the previous iteration is included in the pool. In short, a change of the temporary pool at least indicates the suggested configuration affects the current ensemble. However, though the pool changes, the performance may not be improved due to the greedy mechanism, and thus we count the effective update times (i.e., the pool changes **and the validation error of the ensemble decreases**).
> >
> > As the pool updates very frequently in the beginning, we only calculate the mean effective update times of DivBO, RB-ES, and BO-ES on all datasets during the last 50 and 100 iterations (see the table below). The pool is relatively stable in the last 50 iterations, which also indicates a budget of 250 iterations is sufficient for the datasets. We observe that, on average, DivBO will improve the temporary pool more than once in the last 50 iterations. While the difference between BO-ES and DivBO is the diversity part (also see Q4), we attribute this frequency gain to the use of diversity during the search process.
> >
> > |   | BO-ES | RB-ES | DivBO |
> > | - | :-: | :-: | :-: |
> > | Counts (last 100)   |  1.8  | 2.1 | 3.4 |
> > | Counts (last 50)   |  0.6  | 0.8 | 1.5 |
> >
> > #### Diversity analysis of the final ensemble
> > In addition, we also apply the predictive disagreement[1] to further analyze the diversity of the ensemble. Given two learners, the pair-wise predictive disagreement computes the ratio of disagreed instances for all instances. (Disagreement happens when a learner classifies correctly but the other does not.) The disagreement for an ensemble is computed by averaging the pair-wise disagreement given all pairs of learners in the ensemble. Generally, the larger disagreement is, the more diverse the learners in the ensemble are. We compute the disagreement at the 250-th iteration in Figure 5, and the mean results are as follows,
> >
> > |   | BO-ES | RB-ES | DivBO |
> > | - | :-: | :-: | :-: |
> > | Disagreement     |  0.09  |  0.13  | 0.27 |
> >
> > While the average performance of base learners is similar, the final ensemble built by DivBO enjoys a higher disagreement value (diversity), thus achieving better test accuracy.
> >
> > [1] Tang, E. Ke, Ponnuthurai N. Suganthan, and Xin Yao. "An analysis of diversity measures." Machine learning 65.1 (2006): 247-271.
> >
> > ### Q1. Ablation study.
> > As suggested, we add the results by setting $w=0.1$. The results on five datasets are as follows,
> >
> > | Test Errors (%)  | elevators | house_8L | pol | quake | wind |
> > | - | :-: | :-: | :-: | :-:| :-: |
> > | DivBO (fixed)     |  9.59±0.30 |  11.51±0.28  | 1.66±0.19 | 45.63±1.45 | 14.27±0.36 |
> > | DivBO      |  **9.40±0.28**  |  **10.80±0.22**  | **1.34±0.17** | 45.55±1.37 | **13.93±0.42** |
> >
> > The results show that DivBO with weight schedule (Equation 5) performs much better than fixing the weight for diversity (not significant on quake, but significant on other 4 datasets). It fits the intuition that motivates the weight schedule design in Section 3.2. We will add the ablation study to the experiments.

---

> > > ### Author Response · Authors · 2022-08-02
> > > **Response to Reviewer t73G, Part 3**
> > >
> > > ### Q2. Re-organization.
> > > Thanks for the suggestion. We will remove some details of how we fit the diversity surrogate to the appendix. And as suggested, we will add a sensitivity check in the main paper (move from the appendix) and analyze the effects of $\beta$ and $\tau$. In addition, we will add more experiments, including the significance test (see W3 above) and experiments related to diversity (see W4 above).
> > >
> > >
> > > ### Q3. Explanation of the acquisition function.
> > > The main reason we design the acquisition function is to utilize the variance of the diversity surrogate. If we use a minimum of per-theta-sample-means as suggested, we are approximately comparing the mean value of $M_{div}(\theta, x)$ but ignore the variance, and the most similar learner is the one with the largest mean diversity value. If we use a mean over minimum-of-per-theta as used in DivBO, a base learner from the temporary pool that has low diversity mean but high variance might also be competitive, and this method highlights the uncertainty of the diversity surrogate.
> > >
> > > A similar sampling procedure is applied in an HPO transfer learning work RGPE[1], where they also encourage surrogates with high predictive variance to be sampled. This design may not influence the experiments a lot, but matches the intuition that the uncertainty of the diversity surrogate should also be considered.
> > >
> > > [1] Feurer, Matthias, Benjamin Letham, and Eytan Bakshy. Scalable meta-learning for bayesian optimization using ranking-weighted gaussian process ensembles. AutoML Workshop at ICML. Vol. 7. 2018.
> > >
> > > ### Q4. Explanation of baselines.
> > > BO is exactly DivBO where w equals 0. More precisely, DivBO is an extension of BO by additionally considering diversity when choosing the next configuration to evaluate.
> > >
> > > ### Q5. Explanation of surrogate fitting.
> > > The correlation of the diversity surrogate is still increasing after 250 iterations. Concretely, at the 300-th iteration, the correlation of LightGBM surrogate reaches 0.69 and 0.62 on quake and wind, respectively. The reasons why the correlation is not much higher may be: 1) The ground-truth diversity observations are noisy. Training models based on a given configuration is not deterministic, i.e., the prediction of an instance may be different when training twice based on the same configuration. 2) The Kendall Tau correlation in Figure 2 is computed based on **the predictive means** and the ground-truth diversity observations. Though using predictive means in the experiment can show the effectiveness of the diversity surrogate, it ignores the predictive variance, and the predictive mean may not be precise when the diversity prediction of a configuration pair is of high variance. We will add the above analysis to the experiments.
> > >
> > > ### Q6. Explanation of diversity function.
> > > Thanks for pointing out the problem. In DivBO, we convert the predicted labels of SVCs to class probability (like [1, 0, …]) to prevent errors. We agree that new diversity functions can be applied to support those types of algorithms better, and we consider it a limitation of the current version of DivBO. (See the limitation part below)
> > >
> > > ### Q7. Explanation of Figure 6.
> > > The ensemble selection applies a selection procedure with replacement (as shown in Appendix A.1), which means that some of the learners selected for the final ensemble may be duplicates. We demonstrate the diversity of unique base learners so that the number of base learners seems different.
> > >
> > > ### L1. Limitation.
> > > To clarify the limitation, we will add a paragraph in Section 4.4.  The updated limitations will be as follows,
> > >
> > > Limitation. a) The use of ensemble leads to higher inference latency than using the single best learner (approximately K times where K is the number of learners in the ensemble). This latency can be reduced with the aid of parallel computing if we have sufficient computational resources. In addition, as ensemble selection is fitted on the validation set, there's a risk of overfitting on the test set for small datasets. b) DivBO using Equation 3 as the diversity function can not directly support algorithms that only predict class labels (e.g., SVC). Though DivBO still works by converting the predicted labels to class probability (like [1, 0, …]), other diversity functions can be developed to support those algorithms better.

---

> > > > ### Comment · Reviewer_t73G · 2022-08-05
> > > > **Thank you for the response and the additional empirical evaluations!**
> > > >
> > > > Thank you for the new results and the breakdowns of the original results. I really appreciate the detailed explanations for my different questions regarding the acquisition function and the surrogate fitting.
> > > >
> > > >
> > > > I had a couple of follow-up questions:
> > > >
> > > > - So if I understand correctly, the definition of diversity is not novel but rather from Zhang et al. (2020). Is that a correct assessment?
> > > >
> > > > - For the B/S/W (or Wins/Ties/Losses) comparison, how is the Wilcoxon signed rank test computed between two schemes on the same dataset? Usually, we would compare two schemes across datasets to compute the statistical significance of the difference between the two and/or the direction of the difference. Moreover, is this a one-sided or two-sided test? Also, are these B/S/W numbers corresponding to the numbers in Table 3 of the original paper?
> > > >
> > > > - Thank you for the very nice ablation study on the weight scheduling.  This seems to show that the weight-scheduling is critical to being able to outperform existing non-diverse baselines. For the 5 numbers posted DivBO with fixed weight, in some cases, DivBO has worse performance than all ensembling baseline. Given how critical the weight scheduling is for the DivBO performance, is this weight schedule now a "hyperparameter" that needs to be carefully handled to get improvements over the non-diverse schemes? Or is DivBO robust to the weight scheduling as long as there is a weight scheduling?
> > > >
> > > > - Thank you for the diversity analysis. It is interesting to look at the number of pool updates and the final disagreement in the ensemble. It is a bit counter intuitive that, towards the end of the optimization, when diversity has the highest weight, we are not seeing a larger number of pool updates with diverse candidates -- only 3.4/100. It is higher than the baselines but they are not doing anything to maximize diversity. It might be the case that, towards the end, the algorithm is finding diverse candidates but they are not accurate enough to enter the pool. But that then defeats the point of having diversity aware search. If diversity does not play a large role towards the end, then does the weight schedule seem counter-intuitive? What is the part of the optimization when diversity plays the largest role?
> > > >
> > > > - One final question is that would the baselines like RS-ES or BO-ES be able to catch up with DivBO if they are just given a larger pool of models for the final ensemble selection (for example the whole 250 models attempted during the optimization)? If the pool is larger, there might be (but not necessarily) more diverse models, and they would be selected if they provide any gains.
> > > >
> > > > W. Zhang, J. Jiang, Y. Shao, and B. Cui. Efficient diversity-driven ensemble for deep neural networks. In 2020 IEEE 36th International Conference on Data Engineering (ICDE), pages 73–84. IEEE, 2020.

---

> > > > > ### Author Response · Authors · 2022-08-06
> > > > > **Additional response to Reviewer t73G, Part 1**
> > > > >
> > > > > ### AQ1. Diversity function.
> > > > > Yes. We do not propose a new diversity function but apply the one with potentially good empirical performance from Zhang et al.
> > > > >
> > > > > ### AQ2. Settings of significance test.
> > > > > While the standard deviation is relatively large on some datasets, we think it’s necessary to compare the methods on each dataset. For each dataset, we collect the results of the given two methods across $R=10$ repetitions. And then we conduct the Wilcoxon signed rank test on those pairs (totally $R$ pairs for each dataset). If the difference is not significant, we report an **S** (same). If the difference is significant, and the mean results of DivBO are larger, we report a winning case. If the difference is significant, and the mean results of DivBO are lower, we report a losing case. The setting is similar to that of the significance test in EO \[1\] (see Tables 3 and 5 in EO). Also, the optimization results are the same as those used in Table 3 in our paper. In other words, we conduct the significance test on the optimization results while we present the mean and standard deviation in Table 3 based on the same results. We will clarify the settings in the final manuscript.
> > > > >
> > > > > \[1\] J.-C. Lévesque, C. Gagné, and R. Sabourin. Bayesian hyperparameter optimization for ensemble learning. In Proceedings of the Thirty-Second Conference on Uncertainty in Artificial Intelligence, pages 437–446, 2016.
> > > > >
> > > > > ### AQ3. Parameter sensitivity.
> > > > > We agree that weight scheduling is important for DivBO and sometimes it should be regarded as “hyperparameters”. To analyze how $\beta$ and $\tau$ affect the performance of DivBO, we present the sensitivity analysis. As suggested by Reviewer 15EU, we have improved this part. The sensitivity analysis on two datasets (spambase and House_8L) is as follows,
> > > > >
> > > > > | Spambase | $\tau$=0.05 | $\tau$=0.1 | $\tau$=0.2 | $\tau$=0.4 | $\tau$=0.8 |
> > > > > | - | :-: | :-: | :-: | :-: | :-:|
> > > > > | $\beta$=0.025  |  96.59  |  96.30  | 96.20 |  *96.67*  | 96.34 |
> > > > > | $\beta$=0.05 |  96.12  |  *96.59*  |  **96.78**  |  *96.74*  | 96.41 |
> > > > > | $\beta$=0.1 | 96.01  | 95.98 |  96.27  |  96.23  | 96.30 |
> > > > > | $\beta$=0.2 | 95.76 | 95.80 |  95.80  |  95.76 | 95.68 |
> > > > >
> > > > >
> > > > > | House_8L | $\tau$=0.05 | $\tau$=0.1 | $\tau$=0.2 | $\tau$=0.4 | $\tau$=0.8 |
> > > > > | - | :-: | :-: | :-: | :-: | :-:|
> > > > > | $\beta$=0.025  |  89.60  |  *89.98*  | *89.99* |  89.65  | 90.13 |
> > > > > | $\beta$=0.05 |  89.66  |  89.96  |  *90.10*  |  **90.40**  | *89.95* |
> > > > > | $\beta$=0.1 | 89.96  | 89.76 |  89.25  |  89.83  | *89.98* |
> > > > > | $\beta$=0.2 | 89.37 | 89.40 |  89.60  |  89.34 | 89.17 |
> > > > >
> > > > > Remind that $\beta$ is the maximum of diversity importance and $\tau$ controls the speed of approaching saturation. We observe that a large $\beta$ (0.2) leads to a clear accuracy drop, and we suggest using a $\beta=0.05$. However, we need to tune $\tau$ to achieve the best results on different datasets. The reason may be that the difficulty for different datasets to find good configurations are different. As DivBO builds on the intuition that we need to focus on accuracy rather than diversity in early iterations, a smaller $\tau$ is required if it's difficult to find accurate learners in early iterations. The suggested region for tuning $\tau$ is \[0.1,0.8\]. In our paper, we use 0.2 by default, but a tuned $\tau$ may achieve better results. We will update the sensitivity analysis and add the analysis in Section 3.2.

---

> > > > > > ### Author Response · Authors · 2022-08-06
> > > > > > **Additional response to Reviewer t73G, Part 2**
> > > > > >
> > > > > > ### AQ4. Explanation of weight scheduling function.
> > > > > > We want to note that 3.4/100 is a relatively large update number for BO-based methods. Here, we provide the update number of standard BO **without ensemble**. We count the number of iterations if standard BO finds a configuration with better single-learner accuracy than the best-observed learner. On the same dataset, the number is only 2.3/100. The main reason is that, **the configuration suggested by BO does not guarantee to perform better than previously observed ones, even when the target is to optimize accuracy only.** For example, if the configuration is of high uncertainty, it is also likely to be chosen. Compared with 2.3 in standard BO, the update number of 3.4 is relatively large.
> > > > > >
> > > > > > In addition, we agree that the configurations suggested by DivBO in later iterations have better diversity but worse accuracy compared with BO. As also mentioned by the reviewer, if the accuracy is too low, the configuration may not be selected into the ensemble though it is diverse. To prevent the accuracy from being too low, we highlight the use of $\beta$ in the weight scheduling function, which controls the maximum weight for diversity. While the weight for accuracy is constantly 1, the weight for diversity continuously increases during optimization (**more important given more iterations**). But it increases saturatedly and will not exceed $\beta$. Interestingly, we observe that when $\beta>0.2$, there is a clear accuracy drop (see AQ3 above), which may be the consequence of paying too much attention to diversity as mentioned. By default, we set $\beta$ to be 0.05, so that accuracy also plays an important role while the importance of diversity grows larger. We refer to AQ3 for a detailed analysis of the weight scheduling function.
> > > > > >
> > > > > > ### AQ5. Additional results of larger ensemble size.
> > > > > > The reviewer may suggest analyzing the results if we set a larger ensemble size. As ensemble selection **directly** optimizes the performance on the validation set, **the validation performance is definitely no worse than using a smaller ensemble size** due to the greedy mechanism. However, as pointed out by \[1\], if we optimize the validation set too much (i.e., setting a too large ensemble size for ensemble selection), **the test results may deteriorate**, which is referred to as the overfitting issue in AutoML. The results when setting the ensemble size to 100 for BO-ES are as follows,
> > > > > >
> > > > > > | Test Errors (%)  | elevators | house_8L | pol | quake | wind |
> > > > > > | - | :-: | :-: | :-: | :-:| :-: |
> > > > > > | BO-ES (ens_size=100)    | 9.98±0.30 | 11.52±0.26 | 1.45±0.33 | 47.43±1.62 | 14.04±0.47 |
> > > > > > | BO-ES (ens_size=25)    |  **9.61±0.36**  |  **11.06±0.33**  | **1.35±0.18** | **46.10±2.52** | 14.04±0.53 |
> > > > > >
> > > > > > In our paper, the ensemble size is set to 25 following VolcanoML \[2\], which shows good empirical results across different datasets. We observe that when we set the ensemble size to 100 for BO-ES, the test results are generally worse than setting the ensemble size to 25 due to the overfitting issue (not significant on wind but significant on the other four). We have also mentioned this risk of overfitting in the limitation (see “Response Part3, L1”).
> > > > > >
> > > > > > \[1\] F. Hutter, L. Kotthoff, and J. Vanschoren. Automated machine learning: methods, systems, challenges. Springer Nature, 2019.
> > > > > >
> > > > > > \[2\] Y. Li, Y. Shen, W. Zhang, J. Jiang, B. Ding, Y. Li, J. Zhou, Z. Yang, W. Wu, C. Zhang, et al. Volcanoml: speeding up end-to-end automl via scalable search space decomposition. Proceedings of the VLDB Endowment, 14(11):2167–2176, 2021.

---

> > > > > > > ### Comment · Reviewer_t73G · 2022-08-09
> > > > > > > **Thank you for additional results and responses**
> > > > > > >
> > > > > > > Thank you for the explanations in AQ2 and AQ4 and the new results in AQ3 and AQ5. AQ5 definitely makes sense.
> > > > > > >
> > > > > > > Follow up on AQ3. It is somewhat surprising to see that we need $\beta$ to be so small otherwise diversity hurts more than it helps. It is somewhat disappointing that the crux of this paper is that diversity helps, but in practice, it appears that only a little bit of diversity helps, otherwise it hurts and is worse than no explicit focus on diversity. That is somewhat of a disappointing outcome. While the authors mention that $\tau$ might need to be tuned, the results in the above tables seem to indicate that changes for any given $\beta$ is within a 1%, especially for small values of $\beta$ (which is the recommended value).
> > > > > > >
> > > > > > > However, this also highlights the amount of work needed to take advantage of diversity and highlights the contribution of this paper.

---

### Official Review · Reviewer_eDCC · 2022-07-12

**Rating:** 6
**Confidence:** 5
**Soundness:** 3 good
**Presentation:** 4 excellent
**Contribution:** 2 fair

**Summary:**

This works present a method for hyperparameter optimization applied to ensemble construction. The proposed method builds a surrogate function for diversity of pairs of classifiers that is then combined with a predictive performance objective and maximized. Experiments show that the method outperforms other ensemble building approaches in the literature.

**Questions:**

The paper and method description are clear and self-contained. The question is are the contributions significant enough to be published at Neurips. The performance improvement is small between DiVBO and RB-ES but the experiment compares a wide number of approaches and showcases RB-ES itself I believe for the first time. Find some questions/suggestions below.

- I don't think showing the ranking of validation errors on Figure 3 is very informative, especially given the small size of some of the datasets -- overfitting is likely happening. You should show the ranking of testing error and push back the validation plot to the appendix. Same thing goes for Figure 4 (or maybe reduce number of methods displayed, prune the worse performing methods such as EO, NES and RS-ES and show both validation and testing errors).

- lines 177-179: are the configurations sampled once per objective function? how many points are sampled?

- line 235: we slightly modifies -> modify

- I think section titles 4.2 and 4.3 should be evaluation OF DivBO/Diversity surrogate and not evaluation on?

**Limitations:**

There is a limited discussion of limitations of the method. There are no potential negative societal impacts warranting a discussion.


**Strengths And Weaknesses:**

Strengths:
- The contributions are incremental, inscribed in the already existing framework of ensemble learning through hyperparameter optimization
- The paper is well written and structured
- The quality of the experiments is good

Weaknesses:
- The improvement in performance in comparison with the second best method RB-ES is hinting towards a plateau in performance in terms of ensembling

---

> ### Author Response · Authors · 2022-08-02
> **Response to Reviewer eDCC**
>
> Thanks for your constructive feedback! We believe that addressing this feedback will make our paper significantly stronger. The detailed response to each question is as follows,
>
> ### W1. Significance test.
> To check whether the improvement of DivBO is statistically significant, we conduct the Wilcoxon signed-rank test on each dataset given two methods. The difference is significant when the value $p\leq0.05$ \[1\]. We count the number of datasets if 1) DivBO is better than the other method, and the difference is statistically significant (**B**); 2) the difference is not statistically significant (**S**); and 3) DivBO is worse, and the difference is statistically significant (**W**). For each dataset, the rank of DivBO is 1 for **B**, 1.5 for **S**, and 2 for **W**. We compute the pair-wise rank by averaging the rank on 15 datasets. The results are presented as follows,
>
> **DivBO vs. RB-ES.** The pair-wise ranks of DivBO and RB-ES on all datasets are 1.33 and 1.67, respectively. We agree that RB-ES is a strong baseline. Through the significance test, we observe that DivBO performs no worse than RB-ES on 12 datasets and better on 8 datasets.
>
>
> |   | B | S | W |
> | - | :-: | :-: | :-: |
> | DivBO     |  8  |  4  |  3  |
>
> **DivBO vs. BO-ES.** The pair-wise ranks of DivBO and BO-ES on all datasets are 1.17 and 1.83, respectively. While the DivBO framework is extended from BO, DivBO generally performs better than BO. Concretely, DivBO performs no worse than BO-ES on 13 datasets and better on 12 datasets.
>
>
> |   | B | S | W |
> | - | :-: | :-: | :-: |
> | DivBO     |  12  |  1  | 2 |
>
> **DivBO vs. RS-ES.** The pair-wise ranks of DivBO and RS-ES on all datasets are 1.07 and 1.93, respectively. DivBO performs no worse than RS-ES on all datasets.
>
>
> |   | B | S | W |
> | - | :-: | :-: | :-: |
> | DivBO     |  13  |  2  | 0 |
>
>
> \[1\] Wilcoxon, Frank. Individual comparisons by ranking methods. Breakthroughs in statistics. Springer, New York, NY, 1992. 196-202.
>
> ### Q1. Test results.
> We update the test ranking of four methods on 15 datasets as follows (Each column refers to the mean test rank of the ensemble at that iteration),
>
> |   | 25 | 50 | 75 | 100 | 125 | 150 | 175 | 200 | 225 | 250 |
> | - | :-: | :-: | :-: | :-: | :-: | :-: | :-: | :-: | :-: | :-: |
> | RS-ES    |  2.67  |  3.30  | 3.20 | 3.33 | 3.33 | 3.40 | 3.07 | 3.23 | 3.10 | 3.10 |
> | BO-ES    |  2.53  |  2.37  | 2.77 | 2.67 | 2.73 | 2.60 | 2.93 | 2.87 | 2.80 | 2.80 |
> | RB-ES    |  *2.40*  |  2.20  | 2.10 | 2.13 | 2.13 | 2.30 | **1.97** | 2.13 | 2.10 | 2.27 |
> | DivBO    |  *2.40*  |  **2.13**  | **1.93** | **1.87** | **1.80** | **1.70** | 2.03 | **1.77** | **2.00** | **1.83** |
>
> We observe that DivBO generally achieves the lowest test ensemble error. However, while some datasets are relatively small, building an ensemble on validation set may suffer from a risk of overfitting. At the 175-th iteration, RB-ES performs better than DivBO. To verify whether the improvement at the 250-th iteration is statistically significant, we add a pair-wise significance test (see W1). We will also add the risk of overfitting in the limitation (see 'L1. Limitations' below).
>
> ### Q2. Settings for sampling candidate configurations.
> The configurations to compute the acquisition function are sampled during each iteration. In other words, we need to sample candidate configurations if we want to suggest a new configuration. We sample 1950 configurations randomly from the entire search space and 50 configurations by randomly altering one hyperparameter in the optimal observed configuration. This strategy is also applied in auto-sklearn and VolcanoML. The description is also provided in Appendix A.3.
>
> ### Q3, Q4 Typos.
> Thanks. We will correct the typos.
>
>
> ### L1. Limitations.
> To clarify the limitation, we will add a paragraph in Section 4.4.  The updated limitations will be as follows,
>
> Limitation. a) The use of ensemble leads to higher inference latency than using the single best learner (approximately K times where K is the number of learners in the ensemble). This latency can be reduced with the aid of parallel computing if we have sufficient computational resources. In addition, as ensemble selection is fitted on the validation set, there's a risk of overfitting on the test set for small datasets. b) DivBO using Equation 3 as the diversity function can not directly support algorithms that only predict class labels (e.g., SVC). Though DivBO still works by converting the predicted labels to class probability (like [1, 0, …]), other diversity functions can be developed to support those algorithms better.

---

> > ### Comment · Reviewer_eDCC · 2022-08-05
> > **extensive rebuttal**
> >
> > I would like to thank the authors for their impressive rebuttal. I have read your rebuttal, the other reviews and your answers to those.
> >
> > Thanks for producing the ranking by test accuracy / error, this is a much better metric (actually the only valid metric). I hope you will replace the lines in Figure 3 by the test rankings.
> >
> > I will maintain my score.

---

### Meta-Review · Area_Chair_vcX6 · 2022-08-29

**Recommendation:** Accept
**Confidence:** Certain

**Metareview:**

After a thorough discussion with the authors, all reviewers agree that the paper should be accepted at NeurIPS. The reviewers appreciated the idea of incorporating diversity in the combined algorithm selection and hyper-parameter optimization (CASH) framework and the subsequent use of the diverse models in an ensemble to improve performance. The paper is very clearly written, and the experimental evaluation shows that the proposed techniques provides small but consistent improvements in performance. The authors provided a comprehensive response where they addressed most of the reviewers' concerns. I expect the authors will incorporate all the new results in the camera ready version of the paper.

**Award:**

No

---

### Decision · Program_Chairs · 2022-09-14

Accept